# *NPRL2* gene therapy induces effective antitumor immunity in *KRAS/STK11* mutant anti-PD1 resistant metastatic non-small cell lung cancer (NSCLC) in a humanized mouse model

Ismail M Meraz[1]*, Mourad Majidi[1], Renduo Song[1], Feng Meng[1], Lihui Gao[1], Qi Wang[2], Jing Wang[2], Elizabeth J Shpall[3], Jack A Roth[1]

[1]Department of Thoracic and Cardiovascular Surgery, The University of Texas MD Anderson Cancer Center, Houston, Houston, United States; [2]Department of Bioinformatics and Computational Biology, The University of Texas MD Anderson Cancer Center, Houston, Houston, United States; [3]Department of Stem Cell Transplantation, The University of Texas MD Anderson Cancer Center, Houston, Houston, United States

**\*For correspondence:**
imeraz@mdanderson.org (IMM);
imeraz@mdanderson.org (IMM)

**Competing interest:** The authors declare that no competing interests exist.

## eLife assessment

This study provides a novel and promising NPRL2 gene therapy for enhanced immunotherapy response in a KRAS/STK11 mutant anti-PD1 resistant metastatic NSCLC humanized mouse model. Overall, the authors presented a large amount of **convincing** in vivo data to demonstrate that NPRL2 gene therapy induces antitumor activity through DC-mediated antigen presentation and cytotoxic immune cell activation. This work will be of interest and **useful** to medical biologists and oncologists in the research field of KRAS-mutant NSCLC.

**Abstract** Expression of *NPRL2/TUSC4*, a tumor-suppressor gene, is reduced in many cancers including NSCLC. Restoration of *NPRL2* induces DNA damage, apoptosis, and cell-cycle arrest. We investigated *NPRL2* antitumor immune responses in aPD1^R/*KRAS/STK11*^mt NSCLC in humanized-mice. Humanized-mice were generated by transplanting fresh human cord blood-derived CD34 stem cells into sub-lethally irradiated NSG mice. Lung-metastases were developed from *KRAS/STK11*^mt/aPD1^R A549 cells and treated with *NPRL2* w/wo pembrolizumab. *NPRL2*-treatment reduced lung metastases significantly, whereas pembrolizumab was ineffective. Antitumor effect was greater in humanized than non-humanized-mice. *NPRL2* + pembrolizumab was not synergistic in *KRAS/STK11*^mt/aPD1^R tumors but was synergistic in *KRAS*^wt/aPD1^S H1299. *NPRL2* also showed a significant antitumor effect on *KRAS*^mt/aPD1^R LLC2 syngeneic-tumors. The antitumor effect was correlated with increased infiltration of human cytotoxic-T, HLA-DR^+DC, CD11c^+DC, and downregulation of myeloid and regulatory-T cells in TME. Antitumor effect was abolished upon in-vivo depletion of CD8-T, macrophages, and CD4-T cells whereas remained unaffected upon NK-cell depletion. A distinctive protein-expression profile was found after *NPRL2* treatment. *IFNγ*, *CD8b*, and *TBX21* associated with T-cell functions were significantly increased, whereas *FOXP3*, *TGFB1/B2*, and *IL-10RA* were strongly inhibited by *NPRL2*. A list of T-cell co-inhibitory molecules was also downregulated. Restoration of *NPRL2* exhibited significantly slower tumor growth in humanized-mice, which was associated with increased presence of human cytotoxic-T, and DC and decreased percentage of Treg, MDSC, and

TAM in TME. *NPRL2*-stable cells showed a substantial increase in colony-formation inhibition and heightened sensitivity to carboplatin. Stable-expression of *NPRL2* resulted in the downregulation of MAPK and AKT-mTOR signaling. Taken-together, *NPRL2* gene-therapy induces antitumor activity on *KRAS/STK11*mt/aPD1R tumors through DC-mediated antigen-presentation and cytotoxic immune-cell activation.

## Introduction

*NPRL2* (Nitrogen Permease Regulator-Like 2) is also known as *TUSC4*, a tumor suppressor gene, located on chromosome 3p21.31 in a cluster with other tumor suppressor genes including *TUSC2/ FUS1* (*Kondo et al., 2001*; *Lerman and Minna, 2000*). Reduced expression or deletion of NPRL2 was found in many cancers including lung, renal, colorectal, glioma, gastric, and hepatocellular carcinoma, and has been closely correlated with poor clinical outcomes (*Liu et al., 2014*; *Pi et al., 2022*; *Huang et al., 2016*; *Tang et al., 2014*; *Otani et al., 2009*; *Senchenko et al., 2010*; *Zabarovsky et al., 2002*). We previously reported that low expression of NPRL2 in distinct types of lung cancers was associated with cisplatin resistance (*Ueda et al., 2006*) and restoration of *NPRL2* in *NPRL2* deficient cells over-comes cisplatin resistance through the activation of the DNA damage checkpoint pathway (*Jayachandran et al., 2010*). *NPRL2* was also shown to induce apoptosis, inhibition of cell proliferation, and cell cycle arrest in many other cancer types (*Jayachandran et al., 2010*; *Liu et al., 2015b*; *Li et al., 2004*). *NPRL2* induces apoptosis and DNA damage in cells with active p53 and inactive p53 by causing cell cycle arrest in G1 and G2/M arrest, respectively (*Ma et al., 2017*).

Immunotherapy with checkpoint blockade has shown clinical benefits for NSCLC in only a subset of patients. The majority of patients do not respond (primary resistance) or develop resistance after initial tumor regression (acquired resistance) (*Sharma et al., 2017*). Checkpoint blockade therapy (ICB) was found to be effective primarily among patients with high programmed death protein ligand 1 (PD-L1) expression and with no oncogenic driver mutations such as *EGFR, ALK, or ROS1, MET, KRAS*, and *STK11/LKB1*. The tumors harboring co-mutation of *KRAS* and *STK11/LKB1* (KL) subtypes are also resistant to ICB treatment and showed worse overall survival (OS) and progression-free survival PFS (PFS) in the clinic (*Shire et al., 2020*). The inactivated *STK11* gene was recently reported to weaken the innate immune responses by epigenetic inhibition of stimulator of IFN genes (*STING*) (*Kitajima et al., 2019*). The tumor microenvironment (TME) of these resistant tumors displayed a distinct immune contexture which is characterized by poor infiltration of cytotoxic immune cells and higher infiltration of myeloid-derived suppressor cells (MDSC) (*Tumeh et al., 2014*). Therefore, the development of novel approaches to modulate the TME leading to increased T cell infiltration and its cytotoxic functions within the TME is of paramount importance to overcome ICB resistance.

To this point, compelling evidence indicates the critical role of *NPRL2* in causing DNA damage and double-strand break (*Ueda et al., 2006*; *Jayachandran et al., 2010*; *Ma et al., 2017*) which can be used to trigger DC activation, antigen presentation, and priming of tumor-specific CD8+ T cells in the TME. The effect of tumor suppressor genes in modulating immune cells in the TME has been poorly investigated. The effect of *NPRL2* gene therapy, a tumor suppressor gene, on innate and adaptive immune cells in tumors has not been previously studied. However, we recently established the role of *TUSC2*, another tumor suppressor gene in the same cluster in chromosome 3p21.31, in activating innate and adaptive immunity. *TUSC2* selectively augments natural killer (NK) cells and cytotoxic T lymphocyte (CTL) activity in peripheral blood and the TME, induces a Th1-mediated immune response, and downregulates MDSC and regulatory T cells (Treg) (*Meraz et al., 2018*). We also showed that *TUSC2* gene therapy synergizes with anti-PD-1 in syngeneic *KRAS* mutant lung cancer mouse models (*Meraz et al., 2018*; *Cao et al., 2017*).

We recently developed an improved humanized mouse model using fresh human umbilical cord blood-derived CD34+ stem cells (*Meraz et al., 2019*). The reconstituted humanized mice have a fully competent human immune system with major functional immune populations, which showed antigen-specific T-cell responses as well as antitumor activity with immune checkpoint blockade therapy. This model provides a unique opportunity for establishing an effective drug screening workflow for immunotherapy.

*NPRL2* gene therapy is effective in overcoming cisplatin resistance in NSCLC, but the antitumor immune response and the effectiveness of *NPRL2* have not been studied on ICB-resistant KL (*KRAS*

& *LKB1/STK11* co-mutant) tumors. In this study, we tested the hypothesis that *NPRL2* which induces apoptosis and DNA damage in tumor cells, would promote a variety of innate and adaptive immune responses and exert a strong antitumor effect in overcoming anti-PD1 resistance in KL mutant tumors. Our data show the robust antitumor activity in a variety of anti-PD1 resistant tumors both in syngeneic and humanized mouse models. The efficacy was associated with the induction of an antitumor immune response. The cellular mechanism of *NPRL2* treatment or permanent restoration of *NPRL2* in tumors showed that the antitumor activity was dependent on the induction of antigen-presenting dendritic cells (DC) and activation of cytotoxic T cells. Altogether, our study shows a novel approach of *NPRL2* tumor suppressor gene therapy in sensitizing immunotherapy-resistant tumors.

## Results

### Restoration of *NPRL2* expression mediates inhibition of colony formation and induction of apoptosis in human NSCLC

The basal level of NPRL2 expression is markedly reduced in most NSCLC cells, as we previously reported (*Ueda et al., 2006*). We validated the baseline expression of NPRL2 in H1299, A549, H358, H1975, and H23 cell lines, which was found to be very low or undetectable. Transient transfection with an *NPRL2* expressing plasmid restored *NPRL2* expression (*Figure 1A*, *Figure 1—figure supplement 1*). NPRL2 expression was downregulated in parental H1975 cells, which have an *EGFR T790M* mutation, and its TKI-resistant isogenic cells H1975-OsiR (osimertinib resistant) (*Figure 1—figure supplement 1*). Similarly, in parental H23 cells, which have a *KRAS*$^{G12C}$ mutation and are sensitive to KRASi (sotorasib) and its sotorasib-acquired resistant counterpart H23AR cells, NPRL2 was significantly reduced, and re-expression of NPRL2 was shown by transient transfection with *NPRL2* plasmids (*Figure 1—figure supplement 1*). The restoration of *NPRL2* in anti-PD1 therapy resistant A549 (*KRAS/STK11* mutant) and moderately anti-PD1 sensitive H1299 (*p53* deleted) cell lines showed that the level of PD-L1 expression was increased strongly only in H1299, whereas no change in PD-L1 expression was found in A549 followed after *NPRL2* transfection (*Figure 1A*). Re-expression of *NPRL2* strongly inhibited colony formation in both H1299 and A549 cells which was statistically significant as compared with their respective empty vector (EV) transfection controls (*Figure 1B*) (p=0.005 in H1299; p<0.05 in A549). The degree of inhibition of colony formation in H1299 was found to be stronger than that of A549 (*Figure 1C*). *NPRL2* also induced apoptosis in NSCLC. Scatter plots showed that after transfection of *NPRL2* in H1299 cells, the percentage of annexin V positive cells was significantly increased as compared with the empty vector-transfected group (37.6% in *NPRL2* vs 14.5% in EV; p<0.05) (*Figure 1D and E*). *NPRL2* also induced strong apoptosis in another anti-PD1 resistant mouse lung cancer LLC2 cells (*Figure 1F*).

### Antitumor effect of *NPRL2* gene therapy on anti-PD1 resistant *KRAS/STK11* mutant NSCLC tumors in humanized and non-humanized mice

*STK11/LKB1* mutation is associated with immune resistance to checkpoint blockade therapy. To investigate the antitumor effect of *NPRL2* gene therapy on *STK11/LKB1* mutant anti-PD1 therapy-resistant tumor, A549 lung metastases were formed by injecting A549 NSCLC cells intravenously into fully humanized and non-humanized NSG mice. Metastases were treated with intravenous injection of *NPRL2* gene-loaded cationic lipid nanoparticles (DOTAP-*NPRL2*) with or without pembrolizumab (anti-PD1). Humanized mice were generated by transplanting fresh human cord blood-derived CD34 stem cells into sub-lethally irradiated NSG mice (*Meraz et al., 2019*). The level of engraftment of human CD45$^+$, CD3$^+$ T, CD19$^+$ B, and NK cells was verified before tumor implantation. Mice harboring >25% human CD45 cells were considered humanized. An experimental strategy for non-humanized mice was shown in *Figure 2A* where mice were treated for 3 wk, 10 d after metastasis development. A strong antitumor effect of *NPRL2* gene therapy was found, which was statistically significant compared with empty vector control treatment (*Figure 2B*; p<0.01). The percentage of change in tumor intensity between pre-and post-treatment was significantly different in the *NPRL2* treated group vs control (*Figure 2C*; p<0.04). Bioluminescence images showed the visual differences in tumor burden in the lung between *NPRL2*-treated and empty vector-treated mice (*Figure 2D*).

The experimental strategy for the humanized mouse model was shown in *Figure 2E*, where A549 lung metastasis was developed in 6 wk post-humanized mice followed by *NPRL2* treatment. A robust

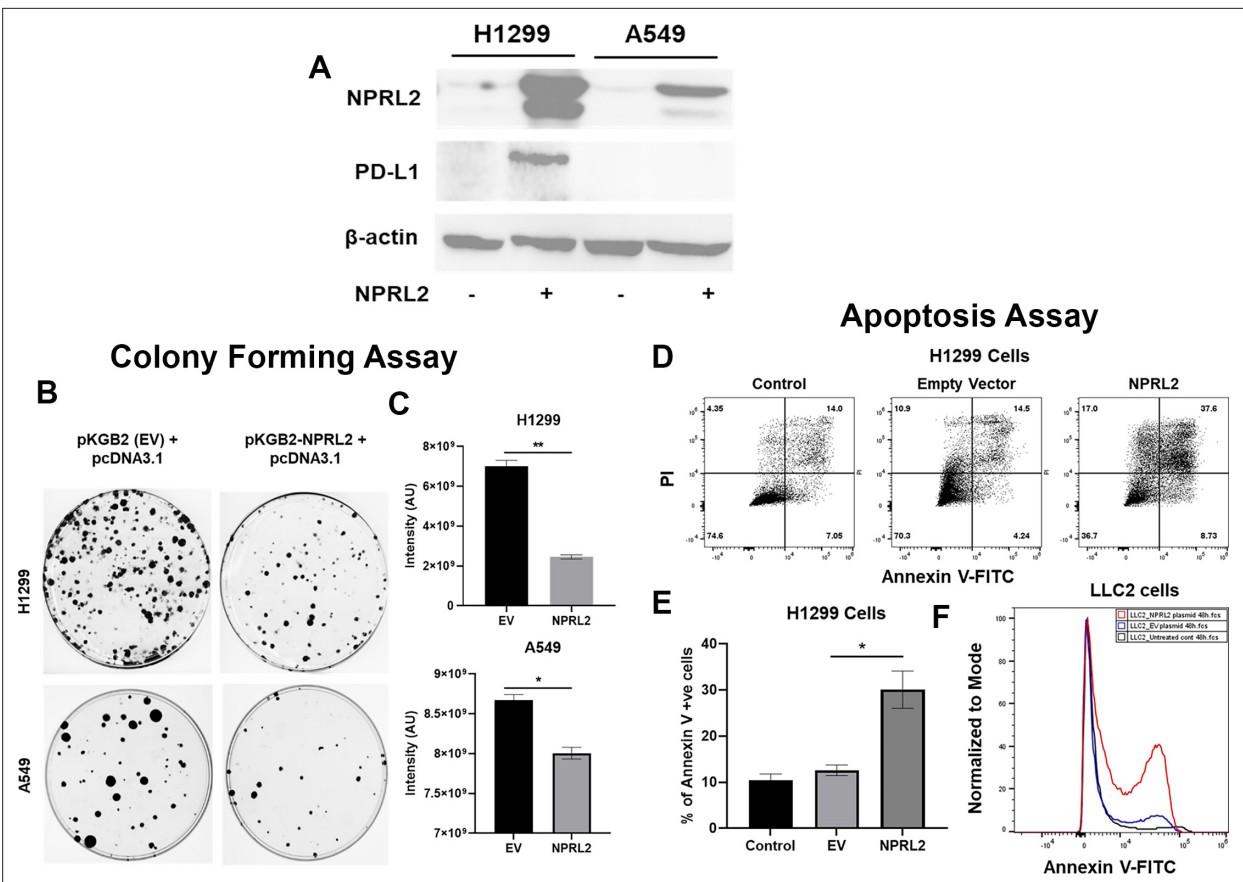

**Figure 1.** Reduced expression of NPRL2 in non-small cell lung cancer (NSCLC) and its restoration caused inhibition of colony formation and apoptosis induction. (**A**) Basal level of NPRL2 and its correspondent death protein ligand 1 (PD-L1) expression were evaluated on anti-PD1 resistant *KRAS/LKB1* mutant A549 and sensitive H1299 cells. (**B**) Inhibition of colony formation of A549 and H1299 after *NPRL2* and respective empty vector transfection and antibiotic selection for 14 d. (**C**) Quantitative measurement of intensity from the colonies that survived after *NPRL2* transfection in A549 and H1299 cells. (**D**) Scatter plots show apoptotic cells in Annexin V and PI double-positive H1299 cells after no transfection, empty vector transfection, and *NPRL2* transfection. (**E**) Percentage of Annexin V positive cells upon *NPRL2* transfection. (**F**) Histogram shows the Annexin V positive LLC2 cells after *NPRL2* transfection. Data is shown as mean percentage ± SD, n=3. *p<0.05; **p<0.005.

The online version of this article includes the following source data and figure supplement(s) for figure 1:

**Source data 1.** Reduced expression of NPRL2 in non-small cell lung cancer (NSCLC) and its restoration caused inhibition of colony formation and apoptosis induction.

**Source data 2.** PDF containing original western blots for *Figure 1A*, indicating the relevant bands and samples.

**Source data 3.** Original files for western blot images displayed in *Figure 1A*.

**Figure supplement 1.** Basal level of NPRL2 expression and its transfection on various NSCLC cell lines.

**Figure supplement 1—source data 1.** PDF file containing original western blots for *Figure 1—figure supplement 1*, indicating the NPRL2 expressions in different NSCLC cell lines.

**Figure supplement 1—source data 2.** Original files for western blot images displayed in *Figure 1—figure supplement 1*.

antitumor activity of *NPRL2* was also recorded in humanized mice, which was significantly different from the control (p<0.01; *Figure 2F*). A significantly low tumor burden was found in the percentage of change in tumor intensity between pre- vs post-treatment in the *NPRL2* treatment group as compared with the empty vector group (p=0.003 *Figure 2G*). When comparing the antitumor activity between humanized and non-humanized mice, the antitumor effect of *NPRL2* was significantly greater in humanized mice as compared with that of non-humanized mice (p=0.008; *Figure 2H*) suggesting that the immune response played a role in inducing antitumor activity. A549 lung metastases were also treated with pembrolizumab (anti-PD1) in combination with *NPRL2* gene therapy in humanized mice. Pembrolizumab did not show any antitumor effect in these anti-PD1-resistant tumors. A dramatic

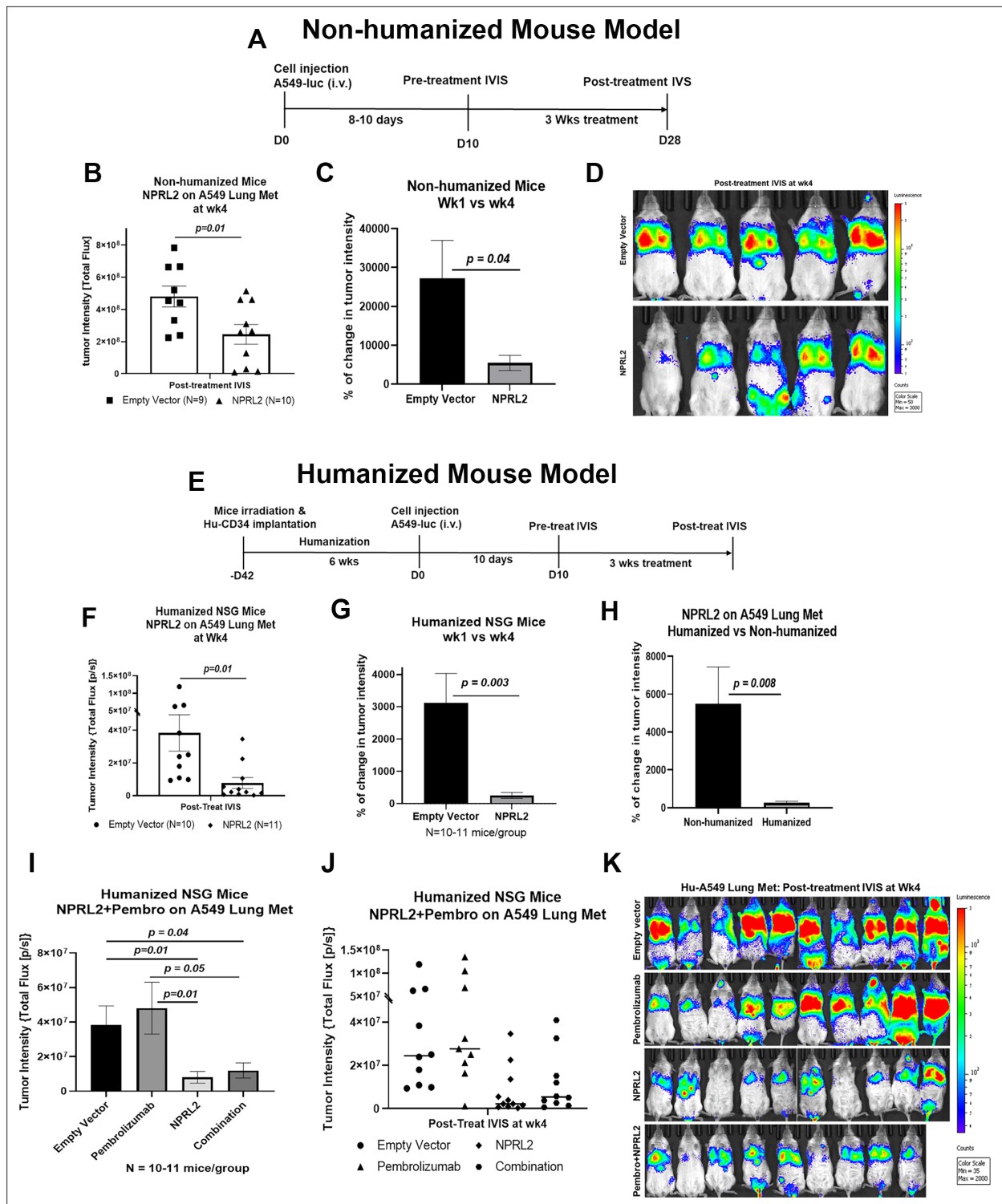

**Figure 2.** Antitumor effect of *NPRL2* gene therapy on anti-PD1 resistant *KRAS/LKB1* mutant tumors in humanized and non-humanized mice. (**A**) Experimental strategy for non-humanized NSG mice shows that tumors were developed through intravenous injection of anti-PD1 resistant A549-luc cells and lung metastasis were treated with *NPRL2* injection i.v. (25 ug/mouse) every 3 d for 3 X (**B**) Tumor-bearing non-humanized mice were imaged by IVIS 200 and tumor intensity was measured and quantitated and level of tumor reduction by *NPRL2* treatment was shown. (**C**) Tumor intensity was measured through bioluminescence imaging and the percentage of change in tumor intensity was calculated by using pre- and post-treatment tumor intensity data (**D**) Bioluminescence imaging was performed by IVIS 200 to visualize the intensity of tumor burden on mice in different treatment groups in non-humanized mouse system. (**E**) Treatment strategy for humanized mice shows that tumors were implanted into 6–7 wk post humanized mice

*Figure 2 continued on next page*

*Figure 2 continued*

and lung metastasis were treated with *NPRL2* injection i.v. (25 ug/mouse) every 3 d for 3 X and pembrolizumab (250 ug/mouse) every 3–4 d for 3 X. (**F**) Tumor-bearing humanized mice were imaged by IVIS 200 and tumor intensity was measured and quantitated and the level of tumor reduction by *NPRL2* treatment was shown. (**G**) Tumor intensity in humanized mice was measured and the percentage of change in tumor intensity was calculated by using pre- and post-treatment tumor intensity data. (**H**) Comparison of the antitumor activity of *NPRL2* between humanized and non-humanized mouse systems based on percentage of tumor intensity change. (**I–J**) Antitumor activity pembrolizumab, *NPRL2* and its combination was determined based on tumor intensity measured through bioluminescence imaging. (**K**) Bioluminescence imaging was performed by IVIS 200 to visualize the intensity of tumor burden on mice in four different treatment groups in the humanized mouse system. This experiment was repeated three times with at least 7–8 mice/group used in each experiment, bars, SD. *$P<0.05$, **$P<0.005$.

The online version of this article includes the following source data for figure 2:

**Source data 1.** Antitumor effect of *NPRL2* gene therapy on anti-PD1 resistant *KRAS/LKB1* mutant tumors in humanized and non-humanized mice.

antitumor effect was mediated by *NPRL2* treatment with or without a pembrolizumab combination (*Figure 2I–J*). No synergistic antitumor activity was found in the *NPRL2* plus pembrolizumab combination. Bioluminescence imaging on mice showed that 7 out of 10 mice contained an extremely low amount of tumor burden in the *NPRL2* treatment group, which was completely and visually different than in the control or pembrolizumab group (*Figure 2K*).

### *NPRL2* mediated antitumor immune responses in immunotherapy-resistant lung metastasis in humanized mice

To investigate the role of *NPRL2* on immune cells in the TME, lung tumor tissues were harvested freshly after the treatment and processed for multicolor flow cytometry. The results showed that infiltration of human CD45[+] cells in tumors was significantly increased in *NPRL2* and *NPRL2* + Pembro combination groups as compared with control (p<0.005 vs control; p<0.05 vs pembrolizumab; *Figure 3A*). Pembrolizumab did not further increase the number of huCD45[+] cells (*Figure 3A*). The percentage of human CD3[+] T (p<0.05 pembrolizumab vs Control; p<0.0005 *NPRL2* vs Control) and cytotoxic T cells (p<0.05 Pembrolizumab vs Control; p<0.005 *NPRL2* vs Control) were increased by both pembrolizumab and *NPRL2* treatment (*Figure 3B–C*), whereas the level of TILs was significantly increased in *NPRL2* treatment as compared with pembrolizumab (p<0.05 vs pembrolizumab; *Figure 3B*). Regulatory T cells were significantly inhibited by *NPRL2* treatment (p<0.005 vs control), whereas the percentage of NK cells was increased by both *NPRL2* alone and combination treatment (*Figure 3D–E*; p<0.0005 control vs *NPRL2*). The number of activated T cells (CD69[+]CD8[+]T) (p<0.005), effector (EM) (p<0.005), and central memory (CM) (p<0.005) CD8[+] T cells were also significantly increased by *NPRL2* treatment (*Figure 3F–*), However, as compared with pembrolizumab, significantly higher number of EM and CM CD8[+] T cells were found after *NPRL2* treatment (*Figure 3H–I*; $P<0.05$). The percentage of tissue-resident memory T cells (CD103[+]CD8[+] T) was also increased by *NPRL2* (*Figure 3J*; $P<0.005$ vs control, p<0.05 vs pembrolizumab). PD-1 expression on CD8[+] T cells was drastically inhibited by pembrolizumab (p<0.00005 vs control) and combination (p<0.0005 vs control) treatment, whereas limited downregulation of PD-1 expression on T cells was seen in *NPRL2* alone treatment (*Figure 3G*). Myeloid cells in tumors are also altered by *NPRL2* treatment. Antigen-presenting HLA-DR[+] DC (p<0.005 vs control; *Figure 3K*) and M1 macrophages (p<0.005 vs control; *Figure 3M*) were significantly increased and CD33[+] MDSC cells were significantly decreased by *NPRL2* (p<0.0005 vs control; *Figure 3L*). Taken together, the antitumor effect of *NPRL2* was associated with increased infiltration of human CD45[+], CD3[+] T, cytotoxic T, NK, and DC cells, and fewer human regulatory T cells (Treg) and MDSC in tumors.

### *NPRL2* induced synergistic antitumor immune response with pembrolizumab on anti-PD1 responsive tumors in humanized mice

As the level of PD-L1 expression was increased with the transfection of *NPRL2* in *Figure 1*, we were interested to see the combinatorial effect of *NPRL2* gene therapy with pembrolizumab immune checkpoint blockade therapy on *KRAS* wild-type H1299 tumors in a humanized mouse model. Seven to eight weeks post-humanized mice were engrafted with H1299 cells followed by treatment. The treatment strategy is shown in *Figure 4A*. The level of humanization was evaluated based on the level of human CD45[+] cells before tumor inoculation (*Figure 4B*) and after treatment (*Figure 4C*). Subcutaneous tumors were developed on humanized mice and the treatment was initiated when the

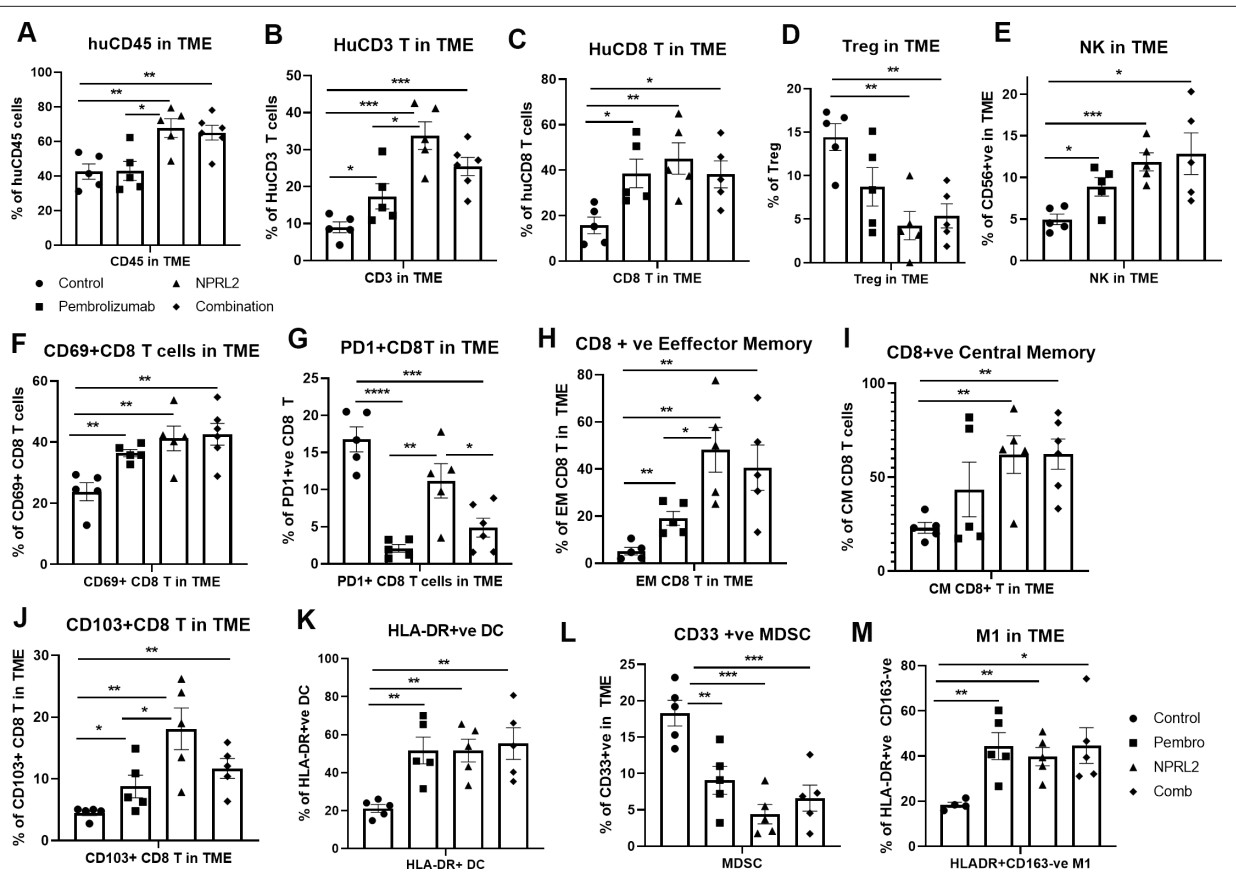

**Figure 3.** *NPRL2* induces antitumor immune responses in anti-PD1 resistant *KRAS/LKB1* mutant A549-Lung Met in humanized mice. A549 lung metastases was developed in 6–8 wk post-humanized mice and lung met-bearing humanized mice were then treated with *NPRL2*, pembrolizumab, and its combination. 3-to 5 d after the treatment, lung metastases tissues were analyzed for infiltrating human immune cells. Single-cell suspensions were prepared from fresh lung metastases and in-depth immune analyses were performed using multicolor flow cytometry for determining human (A) CD45 +leukocytes, (B) CD3 $^+$ T, (C) CD8 $^+$ T, (D) regulatory T cells, and (E) natural killer (NK), cells. The level of human immune cells is shown for different treatment groups. (F) The level of activating CD8 $^+$ T cells was determined by the expression of CD69 expressing markers on infiltrating T cells among different treatment groups. (G) Percentage of PD1 expressing CD8 $^+$ T cells and its alterations after treatment are shown. (H–I) *NPRL2* mediated alteration on the level of effector memory and central memory of CD8 $^+$ T cells in the tumor microenvironment. (J) Percentages of tissue-resident T cells (T$_{RM}$) in tumors and their alteration by *NPRL2* treatment. CD103 $^+$ expressing T cells were considered as T$_{RM}$. (K) The effect of *NPRL2* treatment on the myeloid populations was also investigated. The level of HLA-DR + DC cells was determined among the lineage-negative population. (L) Level of human myeloid-derived suppressor cells (MDSC) based on the expression of CD33 + HLA-DR-ve population and the changes of MDSC in different treatment groups shown in lung met and (M) the percentage of M1 macrophages and its alteration upon *NPRL2* treatment in lung metastases. These populations were gated as Lin-ve >CD11b+ve > HLA-DR+ve > CD163-ve. Statistics are shown at a significance level of p<0.05 unless otherwise noted. Data is shown as mean percentage ± SD, n=5.p<0.05; **p<0.005; ***p<0.0005.

The online version of this article includes the following source data for figure 3:

**Source data 1.** *NPRL2* induces antitumor immune responses in anti-PD1 resistant *KRAS/LKB1* mutant A549-Lung Met in humanized mice: Tumor microenvironment analysis in humanized mice.

tumor size was about 200 mm³. *NPRL2* and pembrolizumab both showed strong antitumor effects, which were statistically significant, whereas a robust and synergistic antitumor effect was observed when *NPRL2* was combined with anti-PD1 immunotherapy (*Figure 4D*). Individual mouse responses to different treatments are shown in *Figure 4E*. Tumor immune microenvironment analysis showed that the percentage of human CD45$^+$ cells at the end of treatment was over 60%, insignificantly different among treatment groups (*Figure 4C*). However, the percentage of cytotoxic CD8$^+$ T and NK cells was significantly increased by the *NPRL2* alone and *NPRL2* +pembrolizumab combination treatment (p<0.05 vs control) (*Figure 4F–G*). An increased number of CD69$^+$ activated CD8$^+$ T cells were found in the pembrolizumab and combination groups (p<0.05 vs control; *Figure 4I*), whereas the level of PD1 expression on CD8$^+$ T was significantly downregulated by pembrolizumab, *NPRL2*, and combination

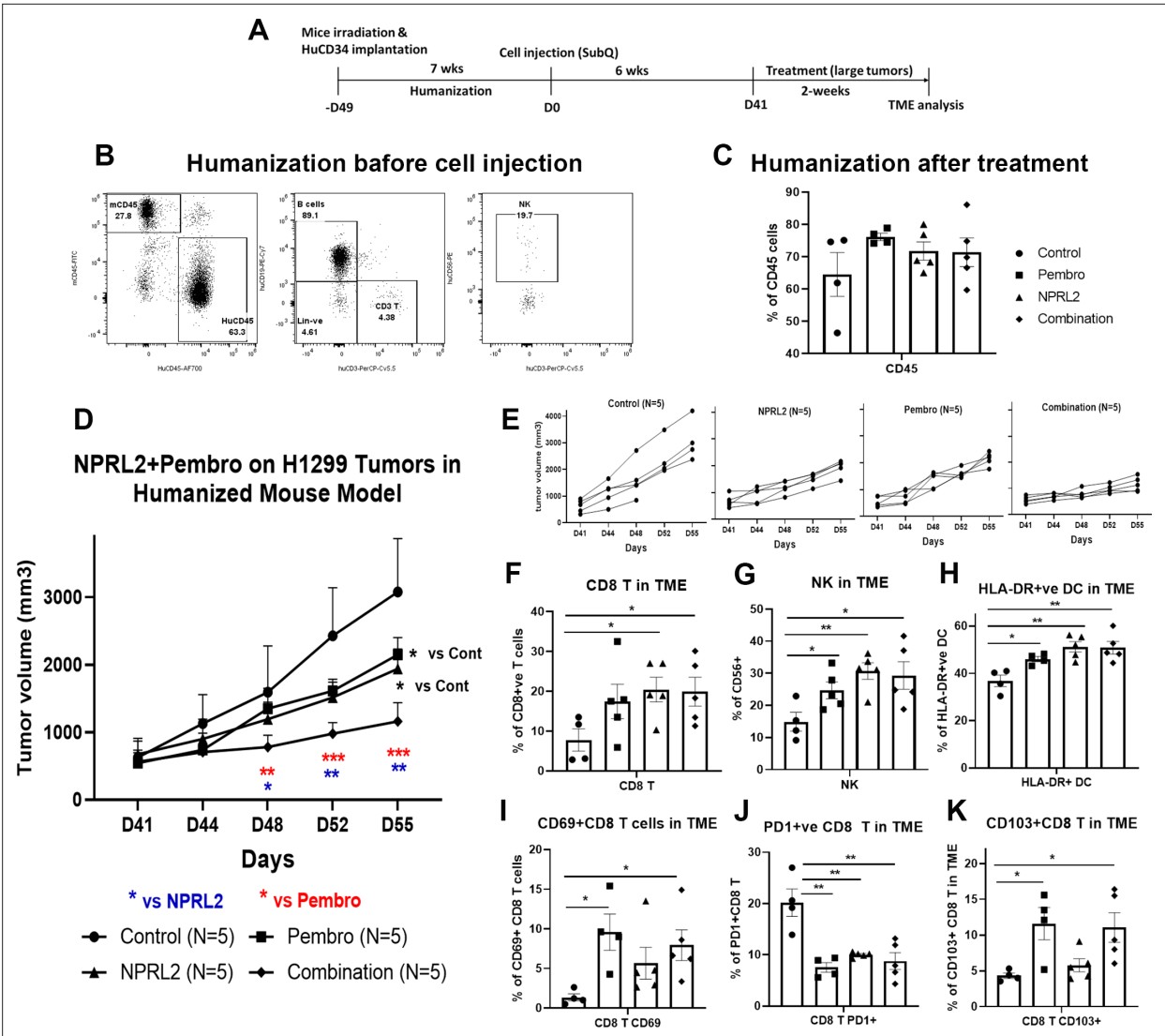

**Figure 4.** *NPRL2* induced synergistic antitumor immune response with pembrolizumab on anti-PD1 responsive H1299 tumors in humanized mice. (**A**) Experimental strategy of H1299 subcutaneous tumor development in humanized mice. NSG mice were humanized for 7–8 wk and humanization was verified by blood screening followed by tumor cell injections. Subcutaneous tumors were developed for another 6 wk to obtain a tumor size of about 200 mm³. Tumors were treated with *NPRL2* (i.v) and pembrolizumab (i.p.) treatment for 2 wk. (**B**) Humanization status was checked by detecting the human CD45 cells in the blood before tumor cell implantation. (**C**) At the end of the experiment, the humanization level was checked based on the number of human CD45 cells in the mouse. (**D**) H1299 tumors were treated with *NPRL2* and pembrolizumab and tumor volume was measured twice a week. The tumor growth curve was generated based on tumor volumes and the antitumor effect was evaluated. (**E**) Growth curves showed the individual mouse response to treatment. (**F–K**) tumor microenvironment analysis was performed to evaluate the immune cell infiltration into tumors. Fresh tumors were harvested within 3–5 d after treatment and single-cell suspensions were prepared for in-depth immune analyses by using multicolor flow cytometry for determining the level of human (F) cytotoxic CD8 $^+$ T, (**G**) effector natural killer (NK) cells, and (**H**) antigen-presenting HLA-DR + DC. (**I**) Level of activating CD8 $^+$ T cells was determined by the expression of CD69 expressing markers on infiltrating T cells among different treatment groups. (**J**) Percentage of PD1 expressing CD8 $^+$ T cells and its alterations after treatment are shown. (**K**) Percentages of tissue-resident T cells (T$_{RM}$) in tumors and their alteration by *NPRL2* treatment. CD103 +expressing T cells were considered as T$_{RM}$. In-vivo experiment was repeated three times with at least 5 mice/group used in each experiment. Statistics are shown at a significance level of p<0.05 unless otherwise noted. Data is shown as mean percentage ± SD, n=5. *p<0.05; **p<0.005; ***p<0.0005.

The online version of this article includes the following source data for figure 4:

**Source data 1.** *NPRL2* induced synergistic antitumor immune response with pembrolizumab on anti-PD1 responsive H1299 tumors in humanized mice: *NPRL2* antitumor effect on H1299 tumors and tumor microenvironment analysis.

treatments (p<0.005 vs control; *Figure 4J*). The number of CD103[+] residential memory T cells was increased in pembrolizumab treatment (p<0.05 vs control), where the number remained unaffected by *NPRL2* (*Figure 4K*). The percentage of HLA-DR[+] DC in the TME was significantly altered by the treatments as compared with control (p<0.05 vs pembrolizumab; p<0.005 vs *NPRL2*; p<0.005 vs comb; *Figure 4H*). Taken together, Cytotoxic T cells, NK cells, and HLA-DR[+] DC were associated with the synergistic antitumor effect of *NPRL2* and pembrolizumab combination in H1299 tumors.

## Antitumor effect of *NPRL2* on anti-PD1 resistant LLC2 tumors in a syngeneic mouse model

LLC2 syngeneic tumors which have an activating $KRAS^{G12C}$ mutation and low PD-L1 expression, are resistant to anti-PD1 checkpoint blockade (*Li et al., 2017*). LLC2 tumors were developed subcutaneously in syngeneic mice and the tumors were treated with *NPRL2*, anti-PD1 therapy, and its combination for 3 wk (*Figure 5A*). Anti-PD1 therapy did not show a significant antitumor effect, whereas a significant tumor reduction was found following treatment with *NPRL2* (p<0.005 vs control; *Figure 5B*). The antitumor effect of *NPRL2* +anti-PD1 was significantly different from control (p<0.05), although this effect is insignificantly different from the *NPRL2* effect alone (*Figure 5B*). Individual mouse growth curve showed that after 3 wk of treatment, *NPRL2* contained a tumor size below 500 mm³ for 50% of mice (4/8), whereas control, anti-PD1, and combination were 0% (0/9), 11% (1/9), and 25% (2/8) (*Figure 5C*). Infiltrating immune cell analysis by multicolor flow cytometry showed that the total number of mouse CD45[+] cells was not significantly altered by the treatments (*Figure 5D*), whereas CD3[+] T, CD4[+]T, and CD8[+]T cells were significantly increased by *NPRL2* treatment (p<0.05 for CD3; p<0.005 for CD4; p<0.005 for CD8 T; *Figure 5E*). Cytotoxic T cells were also increased by the *NPRL2* and anti-PD1 combination (p<0.005) as well as anti-PD1 alone, although, with immunotherapy alone, the level of cytotoxic cells was not changed (*Figure 5E*). NK cells were also significantly increased by *NPRL2* (p<0.005; *Figure 5G*), no change in the number of NK cells was found by the anti-PD1 or combination treatments (*Figure 5G*). Regulatory T cells were downregulated by all treatments (*Figure 5F*) whereas the level of PD1 expression on T cells was not downregulated by the anti-PD1 treatment (*Figure 5H*). *NPRL2* and *NPRL2* combined with anti-PD1 significantly downregulated PD1 expressing CD3[+]T (p<0.005) and CD8[+] T (p<0.0005) cells in tumors (*Figure 5H*). *NPRL2* gene therapy also altered the myeloid population in tumors. As *NPRL2* induced apoptosis in tumor cells (*Figure 1*), we were interested to see antigen presentation by DCs and their maturation. The percentages of CD11c[+] DC (p<0.05; *Figure 5I*) and HLA-DR[+] DC (p<0.005; *Figure 5L*) in tumors were significantly increased by *NPRL2*. The level of DCs was also increased by the combination, whereas anti-PD1 alone treatment did not significantly alter the DC percentage in tumors (*Figure 5I and L*). The level of expression of DC maturation markers including CD86 and CD40 were also increased by *NPRL2*, and the combination as compared with control (*Figure 5M*), whereas the total number of lineage negative myeloid cells was downregulated by *NPRL2* treatment which was significant compared with the control and anti-PD1 groups (*Figure 5K*). The MDSC level was decreased by *NPRL2*, but the most significant reduction in MDSC was found in the anti-PD1 and combination group (*Figure 5J*). Overall, *NPRL2* exerted a significantly strong antitumor effect in a model where anti-PD1 was not effective. Like the A549 NSCLC lung metastasis model, the antitumor effect of *NPRL2* was correlated with the presence of innate immune cells including HLA-DR[+] DC and CD11c[+] DC, and adaptive immune cells including TILs and NK cells in the TME.

## The antitumor immune response of *NPRL2* treatment is dependent on cytotoxic T cells and antigen-presenting cells

To identify the anti-tumor immune cellular mechanism of *NPRL2*, we evaluated the antitumor effect of *NPRL2* on LLC2 syngeneic tumors after depleting cytotoxic CD8[+] T, CD4[+] T, NK, and antigen-presenting macrophages and dendritic cells in mice. The treatment strategy is shown in *Figure 6A*. Antibodies effectively depleted the target-specific CD4[+] T cells from 41 to 0.12%, CD8[+] T cells from 44.2 to 5.29%, and NK cells from 14.6 to 4.9% (*Figure 6B–E*). Clodronate-containing lipid nanoparticles were used to deplete macrophages and DC (*Nguyen et al., 2021*). *NPRL2* showed a strong antitumor effect by inhibiting tumor growth significantly when no immune cells were depleted (*Figure 6B*). When CD4[+] T cells were depleted from the mice, the antitumor effect of *NPRL2* was significantly reduced (*Figure 6C*). There was no antitumor effect of *NPRL2* found when CD8[+] T cells

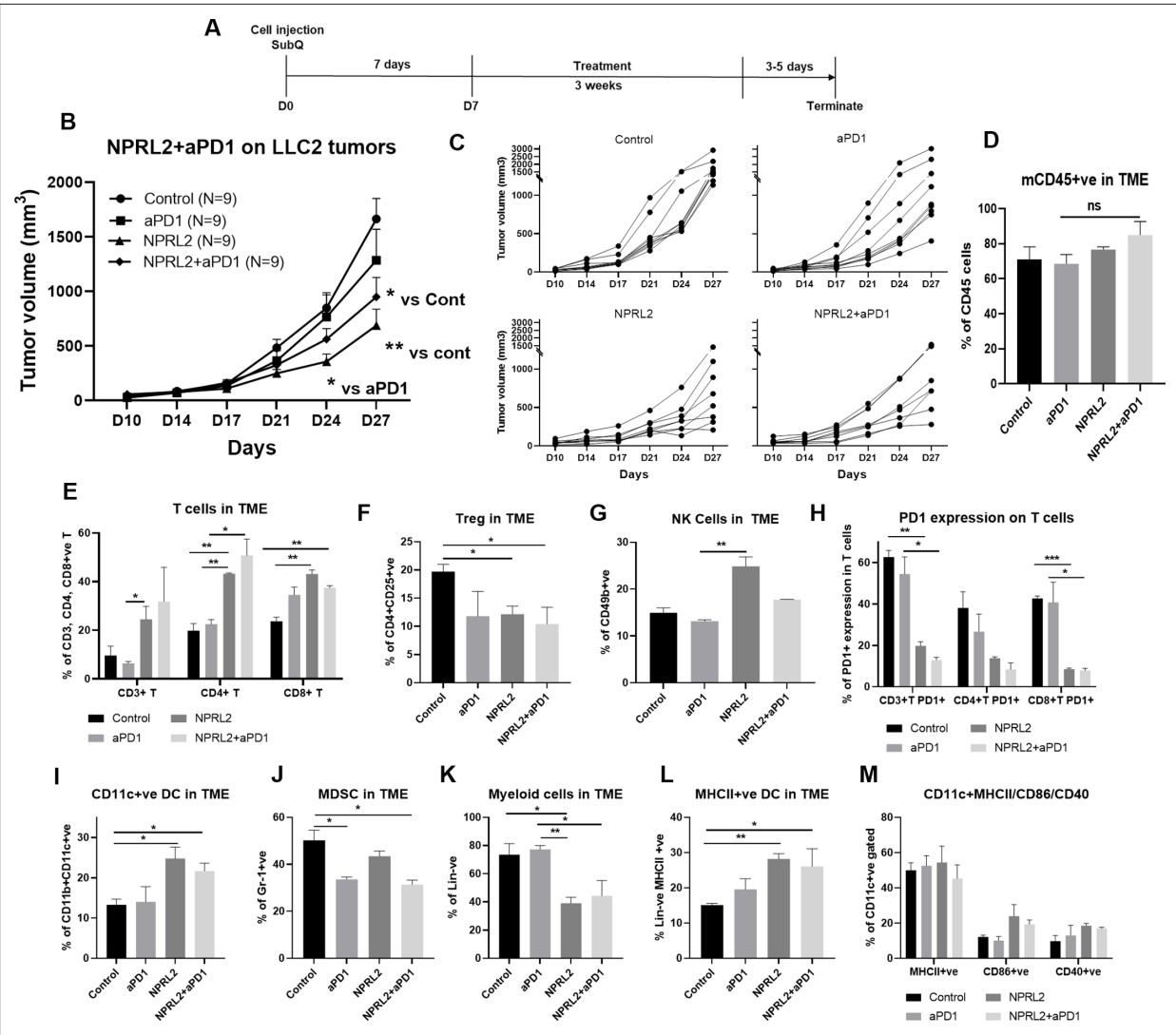

**Figure 5.** Antitumor effect of NPRL2 on anti-PD1 resistant LLC2 tumors in a syngeneic mouse model. (**A**) Treatment strategy of anti-PD1 resistant LLC tumors. Subcutaneous tumors were developed in C57BL/6 mice followed by NPRL2 (i.v.) 25 µg/mouse and anti-PD1 treatments for 3 wk. Tumor volume was measured twice a week. (**B**) The antitumor effect of NPRL2, anti-PD1, and its combinations were shown based on their tumor volume changes for 4 wk. (**C**) Individual mouse growth curves showed the individual mouse response upon treatments. (**D–M**) At the end of the experiment, tumor microenvironment analysis was performed to evaluate the immune cell infiltration into tumors. Fresh tumors were harvested within 3–5 d after the treatment and single cell suspensions were prepared for in-depth immune analyses by using multicolor flow cytometry to determine (**D**) the level of CD45 cells and its alteration due to treatment; (**E**) the percentage of CD3 $^+$ T, CD4 $^+$ T, and CD8 $^+$ T cells in tumors and their changes after NPRL2 treated; (**F–G**) NPRL2 and anti-PD1 treatment effect on regulatory cells and natural killer (NK) cells; (**H**) the level of PD1 expressing CD3 $^+$ T cells (CD3 +CD274+), CD4 $^+$ T cells (CD4 +CD274+) and CD8 $^+$ T cells (CD8 +CD274+) in tumors treated with NPRL2, anti-PD1 and its combination; (**I–M**) Myeloid cells were analyzed after gating out lineage positive cells to see the infiltration of (I) CD11C+DC, (J) MDSC, (K) total myeloid cells, (L) MHCII +DC. (**M**) The maturation status of DC was measured based on the expression of CD40, CD86, and MHCII in CD11c+DC cells. In-vivo experiments were repeated three times with at least 7–8 mice/group used in each experiment. Statistics were shown at a significance level of p<0.05 unless otherwise noted. Data is shown as mean percentage ± SD, n=5. *p<0.05; **p<0.005; ***p<0.0005.

The online version of this article includes the following source data for figure 5:

**Source data 1.** Antitumor effect of *NPRL2* on anti-PD1 resistant LLC2 tumors in a syngeneic mouse model and their tumor microenvironment analysis.

were depleted indicating CD4$^+$ T and CD8$^+$ T cells are important for mediating the antitumor effect of *NPRL2* (***Figure 6D***). *NPRL2* treatment is still effective in inhibiting tumor growth in the absence of NK cells, suggesting NK cells are not essential for the *NPRL2*-mediated antitumor effect (***Figure 6E***). When macrophages were depleted from mice, the antitumor effect of *NPRL2* was also diminished indicating the importance of phagocytic cells for antigen presentation and processing (***Figure 6F***).

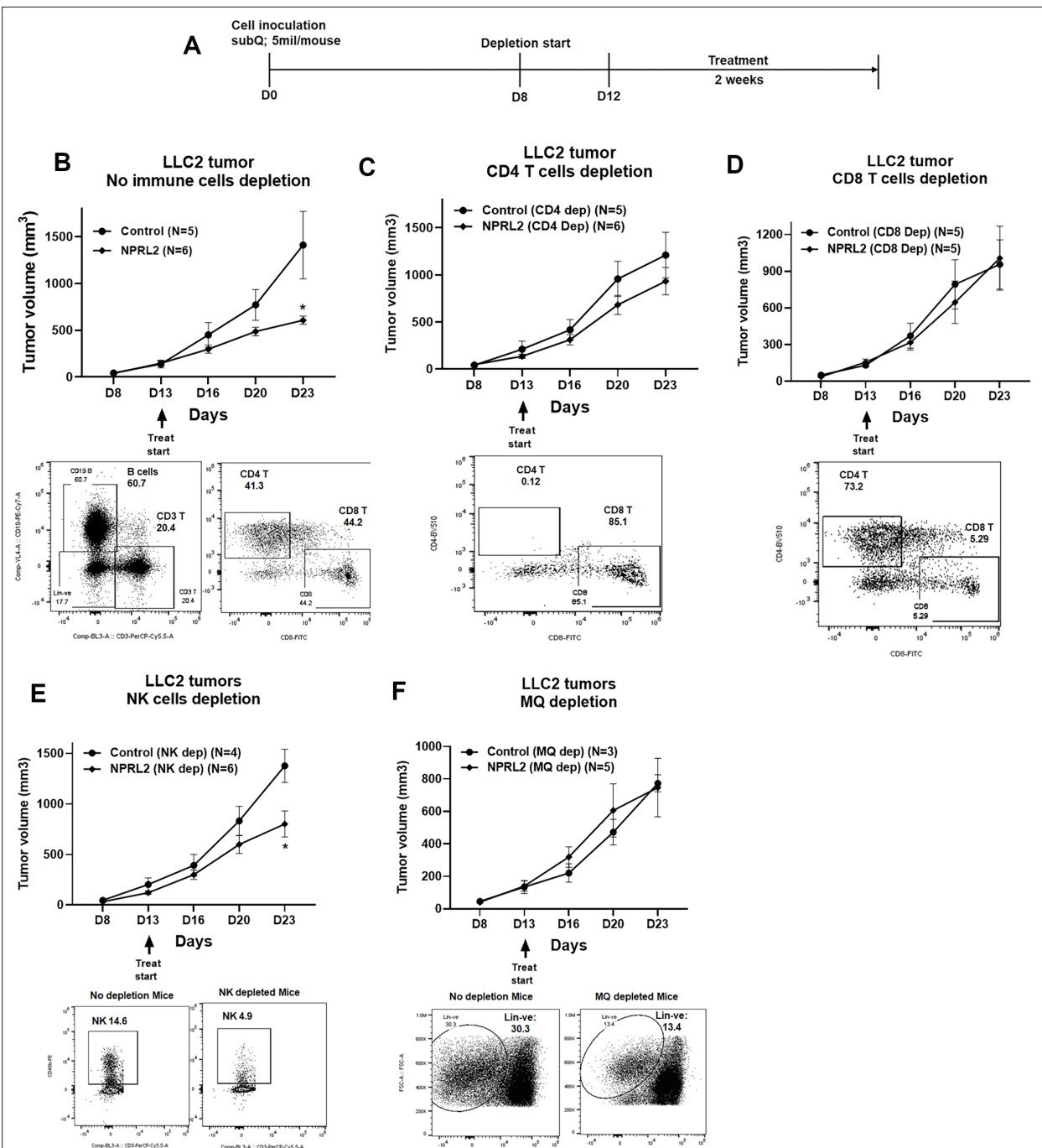

**Figure 6.** Antitumor effect of *NPRL2* on anti-PD1 resistant LLC2 tumors after depletion of immune cells in a syngeneic mouse model. (**A**) Treatment strategy for immune cell depletion experiment. Antibody-mediated CD4 [+] T, CD8 [+] T, natural killer (NK) cell depletion, and clodronate-mediated macrophage and dendritic cell depletion were started 4–5 d before treatment and continued throughout the experiment. (B; top) Antitumor effect of *NPRL2* on LLC tumors on non-depleted C57BL/6 mice which was measured based on tumor volume measurement twice a week for 3 wk; bottom: B cells, CD3 [+] T, CD4 [+] T, and CD8 + T cells status in non-depleted mice. (**C–F**) Antitumor effect of *NPRL2* treatments on LLC2 tumors in CD4 [+] T (C, top), CD8 [+] T (D, top), NK (E, top), and Macrophages/DC (F, top) cells depleted mice. (C-F) Bottom; showed the status of CD4 [+] T, CD8 [+] T, NK, and lineage-negative cells in mice after depleting the cells with respective antibodies and clodronate. The experiment was repeated three times with N=5 mice/group used in each experiment. Statistics were shown at a significance level of p<0.05 unless otherwise noted. Data is shown as mean percentage ± SD, n=5. *p<0.05.

The online version of this article includes the following source data for figure 6:

**Source data 1.** Antitumor effect of *NPRL2* on anti-PD1 resistant LLC2 tumors in immune cells depleted syngeneic mice.

## *NPRL2* increased gene expression associated with T cell activation in tumors in humanized mice

A comprehensive analysis of the expression profile of 770 human immune-related genes was conducted in tumors developed in humanized mice to elucidate the alterations induced by *NPRL2* treatment. A549 lung metastasis resistant to anti-PD-1 was established in fully humanized mice, and *NPRL2* post-treatment samples underwent NanoString analysis. Dendrogram curves (*Figure 7A*) demonstrated a distinct clustering of all *NPRL2*-treated samples, setting them apart from the control samples. Principal Component Analysis (PCA) further supported this observation, revealing a clear separation between *NPRL2*-treated and untreated control samples with over 70% variance (*Figure 7B*). The gene expression heatmap revealed a unique set of genes that were differentially expressed in the *NPRL2*-treated group (*Figure 7C*). Specifically, the expression of genes associated with T cell functions, including *IFNγ, CD8b, CD7, TNFSF18, ITGA1, GATA3, TBX21, CXCR5, IL-12RB1*, and *IL-12A*, was significantly increased upon *NPRL2* treatment (*Figure 7D*). In contrast, genes linked to the negative regulation of T cell functions, such as *FOXP3, TGFB1, TGFB2*, and *IL-10RA*, were markedly inhibited by *NPRL2* (*Figure 7D*). *NPRL2* also significantly downregulated the expression of a set of T cell co-inhibitory molecules including *CTLA, ICOS, LAG3, PDCD1, CD274, IDO1, PDCD1LG2, CD47, KLRB1* (*Figure 7E*). Consistent with the observed upregulation of dendritic cells (DC) in the TME following *NPRL2* treatment, genes related to antigen presentation, including *HLA-DQA1, HLA-DQB1, HLA-DRB4, SLC11A1, SYK*, and *PRKCE*, showed increased expression (*Figure 7F*). The expression of CD68 and CSF1 associated with tumor-associated macrophages (TAM) were significantly downregulated in *NPRL2*-treated samples (*Figure 7F*). Pathway analysis revealed the upregulation of key pathways, including PD-1, PD-L1, PTEN signaling, and PPAR signaling (*Figure 7G*). Network analysis further highlighted increased signaling in IFNγ and IL-4 networks, while TGFβ1 signaling was decreased in *NPRL2*-treated tumors compared to controls (*Figure 7H–J*). These findings collectively underscore the multifaceted impact of *NPRL2* treatment on the immune gene expression profile and signaling networks within the TME.

## Restoration of *NPRL2* expression alters TME in humanized mice

Stably expressing *NPRL2* clones from A549 and H1299 cells were generated, and their impact on tumor growth was assessed in humanized mice. In a comparative study, H1299-*NPRL2*++/++ stable cells and their parental counterparts were implanted into fully humanized NSG mice, and subsequent tumor growth was assessed. The experimental strategy is shown in *Figure 8A*. The humanization status of each mouse was verified at week 5 based on the level of human CD45+, CD3+T, and NK cells before tumor cell inoculation (*Figure 8B*). The results showed that tumors with stable *NPRL2* expression exhibited a notable deceleration in growth compared to the control tumors, as illustrated in *Figure 8C*. Analysis of the TME revealed a distinctive immune context in *NPRL2*-expressing tumors. Notably, there was no significant difference in total human CD45+ cells (*Figure 8D*), but a significant reduction in human B cell numbers was observed in *NPRL2*-expressing tumors (*Figure 8E*). A significant increase in human CD3+ T cells (*Figure 8F*) and cytotoxic T cells were found in *NPRL2*-expressing tumors (*Figure 8G*). No significant effect on human NK cells was found (*Figure 8H*). Further analysis revealed that *NPRL2* expression in tumors significantly inhibited the percentage of regulatory T cells (*Figure 8I*) and PD1-expressing CD3+ T cells (PD1+CD3+ T) (*Figure 8J*), while the levels of CD69+CD3+ T cells (*Figure 8K*) were insignificantly different. The percentage of myeloid-derived suppressor cells (MDSC) (*Figure 8L*) and CD163+ tumor-associated macrophages (TAM) (*Figure 8M*) were markedly downregulated in tumors expressing *NPRL2*, whereas the numbers of HLA-DR+ DC cells were significantly increased (*Figure 8N*). These findings highlight the comprehensive impact of *NPRL2* on the TME, particularly in modulating immune cell populations and associated markers.

## Stable expression of *NPRL2* induces apoptosis and inhibits cell growth by downregulating Akt-mTOR signaling

Retroviral-mediated *NPRL2* stable clones were generated in H1299 and A549 cells, and the expression level of NPRL2 was verified (*Figure 8P*). Subsequent colony formation assays revealed a significant reduction in the number of colonies in *NPRL2* stable clones compared to their counterparts. The inhibitory effect on colony formation was further enhanced with the treatment of the cytotoxic agent carboplatin (*Figure 8O*). In apoptosis assays, while no significant difference in Annexin V-positive cells

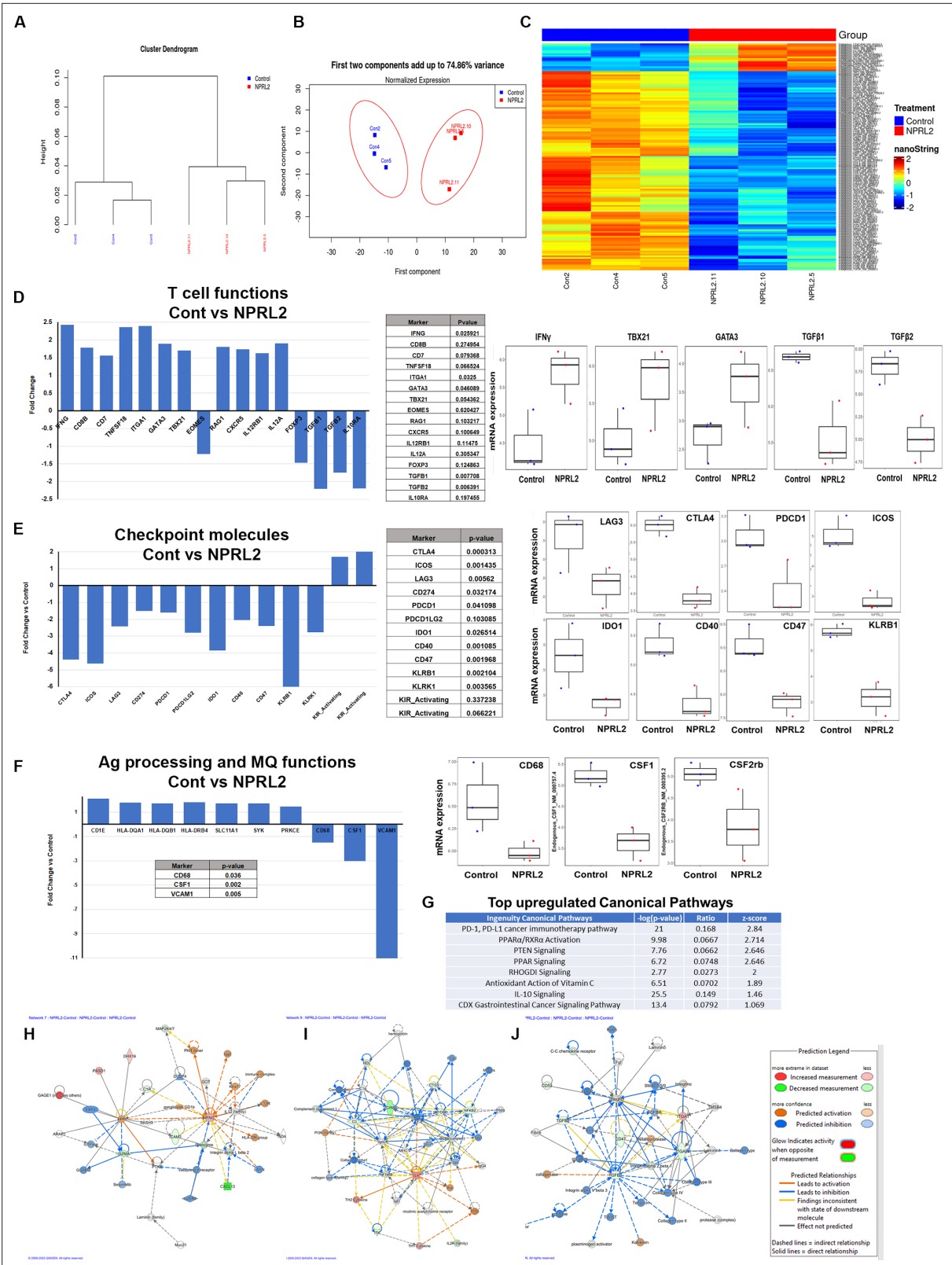

**Figure 7.** Alteration of gene expression and immune signaling associated with T cell activation by *NPRL2* in humanized mice. Expression of 770 human immune-related genes was determined in anti-PD1 resistant lung metastasis in humanized mice by nano string and the effect of *NPRL2* on gene expression was analyzed. (**A**) A hierarchical cluster analysis in the dendrogram showed the distances between clusters; control and *NPRL2*-treated samples. (**B**) The principal of component analysis showed the two components; the first component, and the second component showed the distinct

*Figure 7 continued on next page*

*Figure 7 continued*

clustering of *NPRL2* treated samples from the control group. (**C**) Heat map showing the two sets of genes significantly upregulated and downregulated between two groups. (**D**) The expression of a list of genes involved in T cell functions was upregulated and the expression of another set of genes was downregulated by *NPRL2* treatment in lung metastases. (**E**) The changes in expression of immune checkpoint molecules including co-inhibitory molecules after *NPRL2* treatment. (**F**) The expression of genes associated with antigen presentation and its alteration by *NPRL2* treatment are shown. (**G**) Ingenuity pathway analysis shows a list of top-upregulated signaling pathways upon *NPRL2* gene therapy. (**H–J**) Signaling Network analysis showed three significantly altered immune networks including (**H**) IFNγ, (**I**) IL-4, and (**J**) TGFβ signaling networks by *NPRL2* treatment in lung metastasis; red/brown indicated upregulated, blue/green indicated downregulated, dotted lines mean indirect, solid line means direct. The nano string analysis used three replicates from each treatment group and the data was normalized and statistical analysis was performed by the Department of Bioinformatics.

The online version of this article includes the following source data for figure 7:

**Source data 1.** Alteration of gene expression and immune signaling associated with T cell activation by *NPRL2* in humanized mice: Nano string analysis in tumors developed in humanized mice.

was found between untreated H1299 and H1299-*NPRL2*$^{++/++}$ cells, a significant increase in Annexin V-positive apoptotic cells was observed in carboplatin-treated H1299-*NPRL2*$^{++/++}$ cells compared to parental cells (*Figure 8Q*; left). Similarly, increased apoptosis was also noted in carboplatin-treated A549-*NPRL2*$^{++/++}$ cells, significantly differing from parental A549 cells (*Figure 8Q*; right). The level of apoptosis was verified by the PARP cleavage expression. Both H1299-*NPRL2*$^{++/++}$ and A549-*NPRL2*$^{++/++}$ cells showed higher expression PARP cleavage after carboplatin treatment than that of wild-type H1299 and A549 cells, respectively (*Figure 8R*). Molecular signaling analysis demonstrated significant inhibition of phosphorylation in AKT and MAPK with stable *NPRL2* expression in both H1299 and A549 cells (*Figure 8S*). mTOR phosphorylation was downregulated in A549-*NPRL2*$^{++/++}$ cells vs A549 wild-type cells. Downstream molecules of the AKT-mTOR pathways, including phospho-S6, phospho-4E-BP1, phospho-PRAS40, and phospho-GSK-3b, were consistently downregulated in both *NPRL2* stably expressing cell lines (*Figure 8S*). The permanent restoration of *NPRL2* in cells also exerted an impact on the cell cycle. The number of cells in the G1 phase significantly decreased in H1299-*NPRL2*$^{++/++}$ cells, while an increased number of cells were accumulated in the S and G2/M phases, indicating a significant cell cycle arrest in the G2/M phase (*Figure 8—figure supplement 1*). In summary, these findings collectively suggest that the restoration of *NPRL2* induces apoptosis, inhibits colony formation, alters the cell cycle, and significantly downregulates the AKT-mTOR signaling pathways.

## Discussion

*NPRL2*, a tumor suppressor gene, is downregulated or deleted in many cancers. Our group first identified this gene in a cluster with other tumor suppressor genes in chromosome 3p21.31 (*Lerman and Minna, 2000*). Low expression of NPRL2 is associated with poor clinical outcomes. Restoration of *NPRL2* is multifunctional including inducing apoptosis in cancer cells to inhibit tumor growth. Conversely, the efficient down-regulation of NPRL2 protein expression by both the shRNA and siRNA systems enhanced proliferation, migration, and colony formation in vitro as well as significantly increased tumor growth in hepatocellular carcinoma (*Wang et al., 2022*).

Immune checkpoint blockade (ICB) therapy is showing benefits for only a small percentage of cancer patients. Cancers with high PD-L1 expression and with no oncogenic driver mutations are primarily responsive to ICB. The challenge remains in subtypes of lung cancer patients that harbor driver mutations like *EGFR*, *ALK*, and *ROS1* mutations. Recently, *STK11/LKB1* co-mutated with *KRAS* (KL subtype) were found to be resistant to anti-PD1 treatment due to the presence of a strong association between *STK11/LKB1* genomic alterations and lack of PD-L1 expression on tumor cells (*Jure-Kunkel et al., 2018*; *Skoulidis et al., 2018*). The inactivated *STK11* gene was recently reported to weaken the innate immune responses by epigenetic inhibition of stimulator of IFN genes (*STING*) (*Kitajima et al., 2019*). The TME of these non-responsive tumors is characterized by low infiltration of cytotoxic T cells and enhanced presence of MDSCs. Novel therapeutic approaches are needed to sensitize the ICB-resistant tumors which can strongly modulate the TME.

We have previously shown that *TUSC2* Immunogene therapy, a tumor suppressor gene, can strongly boost the antitumor immune response through a diverse activation of both innate and adaptive immune systems. *TUSC2*-mediated NK cell proliferation enhances the antitumor immune response. Recently, we showed that *TUSC2* as a single agent therapy can alter the TME in KL tumors

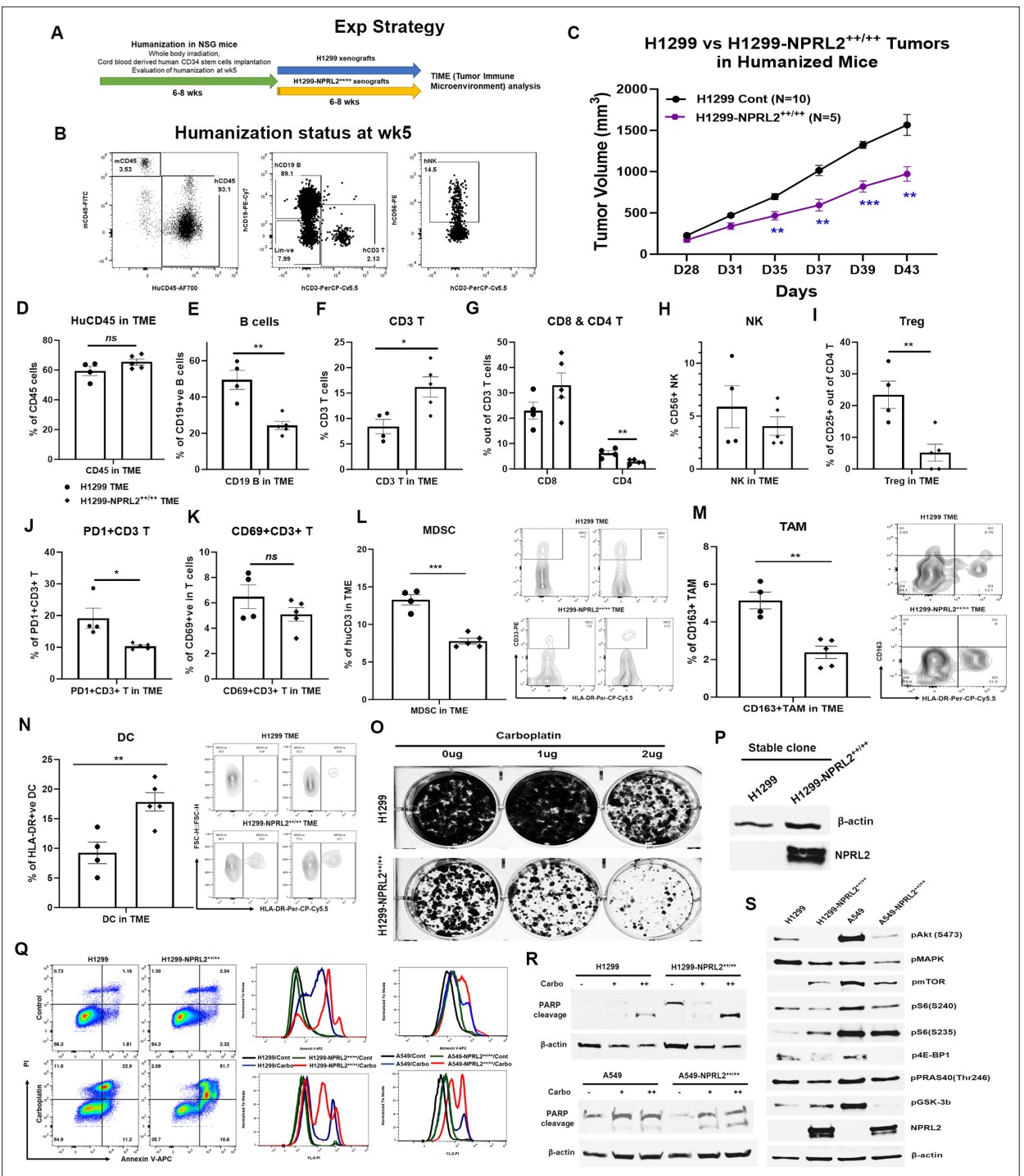

**Figure 8.** Restoration of *NPRL2* expression altered tumor microenvironment (TME), induced apoptosis, inhibited cell growth, and signaling. *NPRL2* stable clones in A549 and H1299 NSCLC cells were generated and developed tumors in humanized mice for tumor microenvironment analysis. The in-vitro assays were performed using these stable clones to elucidate cell death and molecular signaling. (**A**) Scheme showing the experimental strategy where NSG mice were humanized for 6–8 wk followed by H1299-*NPRL2*++/++ tumor cell implantation to develop the tumors for TME analysis. (**B**) Humanization status was checked before tumor implantation based on the level of human CD45 cells. Mice containing 25% or more human CD45 + cells were considered as humanized mice and only the verified mice were used for the tumor implantation. The status of CD3 + T, NK, and B cells was evaluated prior to tumor implantation. (**C**) Tumor growth curves showed the rate of tumor growth and differences in tumor growth between parental H1299 tumors and H1299-*NPRL2*++/++ tumors in humanized mice. (**D–N**) At day 43, fresh tumors were harvested, single cells were prepared, and multicolor flow cytometry was run for multiple innate and adaptive panels to identify the human immune cell populations in the tumor microenvironment. (**D–H**) The percentage of major human immune cells including (D) human CD45+, (E) human CD19 + B, (F) human CD3 + T, (G) CD4 +

*Figure 8 continued on next page*

*Figure 8 continued*

T, and CD8 + T, and (**H**) natural killer (NK) cells in parental tumors and their alterations in *NPRL2*-stably expressing tumors in humanized mice. (**I–K**) The status of (**I**) regulatory T cells, (**J**) PD1 expressing T cells (CD274 +CD8+ T), and (**K**) activating T cells (CD69 +CD8+ T) in both parental and *NPRL2*-stably expressing tumors. (**L–N**) The presence of innate cells was also investigated and analyzed after gating out the lineage-positive population to identify the level of (**L**) MDSC (CD33 +HLA-DR- MDSC), (**M**) TAM (CD11b+HLA-DR-CD163+TAM), and (**N**) HLA-DR +DC in tumors in humanized mice. (**O–S**) A549-*NPRL2*++/++ and H1299-*NPRL2*++/++ cells were used for in-vitro assays compared with their respective parental counterparts. (**O**) Stable expression level of *NPRL2* was verified by western blot, (**P**) Colony forming assay in H1299 and H1299-*NPRL2*++/++ cells showed the differences in colony formation inhibition in the absence or presence of different concentrations of carboplatin, (**Q**) Apoptosis assay was also performed in both pairs of cell lines to detect the annexin V positive apoptotic cells by flow cytometry following carboplatin treatment. The difference in the level of apoptosis was estimated and compared between parental and *NPRL2* stably expressing counterparts. (**R**) The level of apoptosis was verified by PARP cleavage by detecting the cleavage of PARP by western blots in NRPL2 stable cells after carboplatin treatment. (**S**) Western blots were performed to detect a list of signaling molecules involved in downstream and upstream of the PI3K/AKT/mTOR signaling pathway, which included p-AKT, p-mTOR, p-S6, p-4E-BP, p-PRAS40, p-GSK-3b. The MAPK pathway was investigated based on the level of expression of pERK1/2. The in-vivo experiment was repeated three times with at least N=5 mice/group used in each experiment. Statistics were shown at a significance level of p<0.05 unless otherwise noted. Data is shown as mean percentage ± SD, n=5. *p<0.05; **p<0.005; ***p<0.0005.

The online version of this article includes the following source data and figure supplement(s) for figure 8:

**Source data 1.** Restoration of *NPRL2* expression altered tumor microenvironment (TME), induced apoptosis, inhibited cell growth and signaling: Tumor microenvironment analysis in *NPRL2* stable expressing tumors.

**Source data 2.** PDF file containing original western blots for *Figure 8R and S*, indicating relevant bands, treatments and samples.

**Source data 3.** Original files for western blot images displayed in *Figure 8R and S*.

**Figure supplement 1.** Stable expression NPRL2 affected the cell cycle.

**Figure supplement 1—source data 1.** Cell cycle analysis in NPRL2 stably expressing cells.

and enhance the antitumor effect when combined with standard chemo-immunotherapy on the *LKB1/STK11* mutant lung model (*Meraz et al., 2022*). Like *TUSC2*, adenoviral *p53* (Ad-*p53*) gene delivery in combination with ICB and selective agonist overcomes the ICB resistance by increasing CD8+ T cell, antigen presentation signatures and by decreasing immune-suppressive TFG-beta gene signatures (*Chada et al., 2022*).

Previous studies functionally characterized *NPRL2* and its molecular mechanism in cancer cells (*Liu et al., 2015b*; *Bar-Peled et al., 2013*), but the role of *NPRL2* in modulating the TME and its association with immune cells in tumors has not been previously studied. A bioinformatics analysis in stomach adenocarcinoma found a negative correlation between ImmuneScore, StromalScore, and ESTIMATEScore and expression of NPRL2. The study found a negative correlation among STAAD cancer patients between NPRL2 expression and PDCD1LG2, LAIR1, and BTLA checkpoint markers. Co-stimulatory markers TNFSF9 and TNFRSF14 are positively correlated with *NPRL2* expression (*Pi et al., 2022*).

Our study is the first to investigate the antitumor immune response of *NPRL2* on anti-PD1 resistant *STK11* mutant NSCLC in a humanized mouse model. Consistent with our previous studies, *NPRL2* delivery induced apoptosis and inhibited colony formation in an array of anti-PD1 moderately (H1299) or resistant cells (A549, LLC2) as well as inhibited tumor growth in syngeneic, immune deficient, and immune-competent humanized mouse models. The degree of antitumor effectiveness is significantly greater in immune-competent humanized mice systems than immune deficient xenograft tumors suggesting the role of *NPRL2* in antitumor immune responses.

The level of PD-L1 expression was increased with the delivery of *NPRL2* in H1299 cells, and its tumors in humanized mice were synergistically susceptible to the combination of *NPRL2* gene therapy and pembrolizumab (anti-PD1). The level of PD-L1 expression was not increased in *KRAS/STK11* mutant A549 and *KRAS* mutant LLC2 tumors with *NPRL2* delivery, and no synergistic effect was found in *NPRL2* and pembrolizumab combination treatments in humanized mice, although a significant anti-tumor activity was found by *NPRL2* gene therapy in these resistant tumors where pembrolizumab alone is ineffective. This finding suggests that the PD-1/PD-L1 signaling axis is broken in A549 (KL) and K LLC2 (K) tumors and *NPRL2* gene therapy overcomes this resistance independent of PD-L1 expression.

There is a distinct difference in immune contexture between responsive (H1299) and resistant (A549) tumors upon pembrolizumab treatments in humanized mice. In anti-PD1 resistant KL tumors while both pembrolizumab and *NPRL2* treatments increased the CD8+ T cells in TME, only *NPRL2*

treatment increased the presence of effector memory CD8$^+$ T cells and residential memory CD8$^+$ T (CD103$^+$ CD8$^+$ T) in TME which was significantly different than that of pembrolizumab treatment (*Figure 3H and J*). These findings suggest that in anti-PD1 resistant tumors, the CD8$^+$ T cells in TME are not fully functional, although the level of PD1 expression on CD8$^+$ T was significantly decreased by pembrolizumab treatment (*Figure 3G*). On the other hand, in anti-PD1 responsive tumors, pembrolizumab significantly increased activated (CD69$^+$ CD8$^+$ T; PD1$^+$CD8$^+$ T) and memory (CD103$^+$ CD8$^+$ T) T cells (*Figure 4I–K*). Moreover, the percentage of exhausted CD8$^+$ T cells was significantly higher after anti-PD1 treatment in resistant LLC2 tumors (*Figure 5H*). These findings indicate the differences in functional CD8$^+$ T cells in resistant vs sensitive tumors.

The CD8$^+$T cells are activated by helper T cells and innate activation. *NPRL2* gene therapy was effective on both resistant A549 and LLC2 tumors and was associated with an increased number of CD4$^+$ T cells as well as antigen-presenting dendritic cells (DC). CD86 expression was increased in DCs whereas MDSC was significantly downregulated by *NPRL2* treatment. The depletion of CD8$^+$ T, CD4$^+$ T cells, and DCs abolished the antitumor effect of *NPRL2* indicating that *NPRL2* activates innate immunity which leads to activation of CD8$^+$ T cells in resistant tumors. *NPRL2* activity was found to be less dependent on NK cells, although *TUSC2* gene therapy, another tumor suppressor gene, preferentially activates NK cells (*Meraz et al., 2018*).

A wide array of human immune gene signatures and their alteration by gene therapy was investigated in lung metastases in humanized mice by using nano-string technology. This is one of the very few studies where high throughput analysis was performed in a humanized mouse system to see the human immune gene expression profiles (*Kenney et al., 2022*). Consistent with the results found in multicolor flow cytometry, a set of genes associated with the function of CD8$^+$ T cells were significantly upregulated by *NPRL2* in *KRAS/STK11* mutant anti-PD1 resistant tumors. A number of T cell co-inhibitory molecules *CTLA4, ICOS, CD274, PDCD1*, and *IDO* were markedly downregulated by *NPRL2* treatment which was in agreement with the data found in bioinformatics analysis in stomach cancer patients' samples (*Pi et al., 2022*). The gene signatures associated with DC activation and antigen presentation were found to be unregulated by *NPRL2* which was in line with flow cytometry results.

The TME was significantly altered when *NPRL2* was permanently restored, which was characterized by increased pro-immunogenic signatures including increased TILs, CD8$^+$ T, and HLADR$^+$ DC, which led to significantly reduced tumor growth in humanized mice. On the contrary, the knockdown of *NPRL2* promoted cell growth and proliferation in vitro and in vivo (*Wang et al., 2022*). Ingenuity pathway analysis showed the significant downregulation of pro-tumorigenic TGFβ1, TGFβ2, and CD47 signaling in anti-PD1 resistant tumors after *NPRL2* gene therapy. Consistent with this gene expression data, tumor-promoting immune suppressive immune cells Treg, MDSC, and TAM were significantly downregulated in *NPRL2*$^{++/++}$ tumors in humanized mice.

Colony formation was impaired in *NPRL2*$^{++/++}$ cells and this inhibition of colony formation and induction of apoptosis were significantly enhanced with the carboplatin treatment. Our group along with others in different cancers previously reported that NPRL2 enhances sensitivity to chemotherapy and overcomes drug resistance (*Ueda et al., 2006*; *Jayachandran et al., 2010*; *Liu et al., 2015a*; *Liu et al., 2015c*; *Liu et al., 2019*). Other studies also have shown that NPRL2 promotes chemoresistance in prostate cancer by activating autophagy (*Luo et al., 2020*; *Chen et al., 2019*). In addition, consistent with others, *NPRL2*$^{++/++}$ expressing cells were also found to be arrested in the G2/M phase in *p53*-null cells (*Jayachandran et al., 2010*; *Ma et al., 2017*).

Constitutively active AKT signaling has been found in a variety of cancers which is downregulated by *NPRL2* (*Liu et al., 2019*). In this study, phospho-AKT was found to be significantly downregulated in *NPRL2* stably expressing cells as compared with its parental counterparts.

The mTOR complex 1 (mTORC1) pathway promotes cell growth through the Rag GTPases to promote mTORC1 translocation and its activation. mTOR activity is inhibited by NPRL2 by interacting and directly binding with Rag GTPase, RagD in particular, and interferes with mTOR activity (*Bar-Peled et al., 2013*). mTOR was found to be downregulated in *NPRL2*-restored A549-*NPRL2*$^{++/++}$, which was consistent with others (*Liu et al., 2015b*; *Liu et al., 2015c*), whereas knocking down *NPRL2* increased the Rag GTPases and mTOR activation (*Wang et al., 2022*). This mTOR downregulation was found to be associated with decreased expression of downstream mediators including phospho-S6, phospho-4E-BP1, phospho-PRAS40, and phospho-GSK-3β in *NPRL2* restored cells. NPRL2 can also positively regulate the mTOR activity by binding directly to Raptor, instead of Rag GTPase, although

binding affinity to raptor is minimal (*Kwak et al., 2016*). The MARK signaling pathway was also found to be downregulated by *NPRL2* in this study.

In this study for the first time, it has shown that the tumor suppressor gene, *NPRL2*, can induce significant antitumor immune responses by altering the tumor immune microenvironment and overcoming anti-PD1 resistant tumors. Anti-PD1 resistant *KRAS/STK11* subtype TME can be reshaped with increased infiltration of TILs with their cytotoxic function along with its direct effect on apoptosis induction and growth inhibition through downregulating AKT/mTOR and MAPK signaling.

## Materials and methods

### Cell line, cell culture, and maintenance

Anti-PD1 resistant A549-luc human NSCLC cell line which carries *KRAS/STK11* mutation, was kindly provided by Dr. John Minna (The University of Texas Southwestern Medical Center, Dallas, TX). A549-luc cells were cultured in RPMI-1640 medium supplemented with 10% heat-inactivated fetal bovine serum (GE Healthcare Life Sciences, HyClone Laboratories) and 1% penicillin-streptomycin (Thermo Fisher Scientific) at 37 °C with 0% $CO_2$. Another anti-PD1 resistant syngeneic mouse cell line LLC2-luc also carries *KRASG12C* mutation, was kindly provided by Dr. Yonathan Lissanu (MD Anderson Cancer Center). LLC2 cells were cultured in Dulbecco's modified Eagle's medium supplemented with 10% heat-inactivated fetal bovine serum (GE Healthcare Life Sciences, HyClone Laboratories) and 1% penicillin and streptomycin (Thermo Fisher Scientific). H1299-luc human NSCLC cell line harbors *p53*$^{nul}$ and *KRAS* wild-type mutation was obtained from ATCC. All cell lines tested negative for mycoplasma before use in experiments. The cell lines were tested for mycoplasma routinely by ELISA in a core lab at MD Anderson Cancer Center. The cell lines were also authenticated before the experiment by the core lab at The University of Texas MD Anderson Cancer Center.

### Mice were used for in-vivo studies

NOD. Cg-*Prkdc*$^{scid}$ *Il2rg*$^{tm1Wjl}$/SzJ (NSG) mice were obtained from The Jackson Laboratory, which was used for humanization and anti-PD1 resistant and sensitive lung metastasis and subcutaneous xenograft tumor development for immunotherapy research. Female 3-to-4-wk-old mice were used in these studies. Mice were housed in micro isolator cages under specific pathogen-free conditions in a dedicated humanized mice room in the animal facility at The University of Texas MD Anderson Cancer Center. Mice were given autoclaved acidified water and fed a special diet (Uniprim diet). C57BL/6 mice were used for the development of LLC2 syngeneic tumors. These mice were also used for in-vivo depletion of immune cells and evaluation of anti-tumor response on immune cell-depleted mice. All animal experiments were carried out following approval by the MDACC institutional review board and were performed by the Guidelines for the Care and Use of Laboratory Animals published by the National Institutes of Health.

### Generation of humanized mice

The development of humanized mice from fresh human cord blood-derived CD34 stem cells was described previously (*Meraz et al., 2019*). Human umbilical cord blood units for research were obtained from MD Anderson Cord Blood Bank under an Institutional Review Board (IRB)-approved protocol (Lab04-0249). The cord blood bank collects umbilical cord blood through voluntary donations from mothers following informed consent under the institutionally approved IRB protocol. Cord blood bank collects human cord blood on a daily basis from several Houston-area hospitals like Memorial Hermann Hospital, St. Joseph Medical Center, and the Woman's Hospital of Texas, etc. Fresh cord blood units were delivered to the research lab within 24 hr of harvest, and the cord blood units were HLA typed immediately at MD Anderson HLA-typing core facility. Cord blood was diluted to a ratio of 1:3 with phosphate-buffered saline, and mononuclear cells were isolated by using density-gradient centrifugation on the Ficoll medium. The isolated mononuclear cells were directly used for CD34$^+$ enrichment.

After mononuclear cells were separated from human umbilical cord blood, CD34$^+$ HSPCs were isolated using a direct CD34$^+$ MicroBead kit (Miltenyi Biotec). Three- to 4-wk-old NSG mice were irradiated with 200 cGy using a $^{137}$Cs gamma irradiator. Over 90% pure freshly isolated CD34$^+$ HSPCs were injected intravenously, 24 hr after irradiation, at a density of 1–2×10$^5$ CD34$^+$ cells/mouse. The

engraftment levels of human CD45[+] cells were determined in the peripheral blood, as early as 4 wk post CD34 injection, by flow cytometric quantification, as well as other human immune populations. Mice with 25% human CD45[+] cells were considered humanized (Hu-NSG mice). Hu-NSG mice from cord blood donors with various levels of engraftment were randomized into every treatment group in all the experiments. All Hu-NSG mice were verified for humanization before tumor implantation.

## *NPRL2* gene therapy formulation

*NPRL2*, a tumor suppressor gene, was encapsulated into DOTAP nanoparticles. The formulation is composed of 1, 2-bis(oleoyloxy)–3-(trimethylammonio) propane (DOTAP): cholesterol nanoparticles and a DNA plasmid expressing the *NPRL2* tumor suppressor gene. The plasmid contains a kanamycin resistance gene, an origin of replication, and the human wild-type *NPRL2* gene driven by a CMV promoter. The formulation is routinely manufactured in GMP facility following standard manufacturing protocols by Genprex Inc The GMP-grade *NPRL2* nanovesicles were obtained from Genprex Inc and used as a study agent in this study. *NPRL2* encapsulated nanovesicles were validated for their expression by transfecting into A549 and H1299 cells before inoculation in humanized mice.

## Generation of *NPRL2* stable clone

A549-*NPRL2*[++/++] and H1299-*NPRL2*[++/++] stable clones were generated by the MD Anderson core facility through adenovirus-mediated *NPRL2* transduction (OriGene, Rockville, MD) into A549 and H1299 cell lines and the stable clones are selected based on antibiotic treatment and the individual clones were selected through screening of multiple clones by evaluating the level of NPRL2 expression by western blot.

## Colony formation assay

To analyze the effect of NPRL2 protein on inhibition of colony formation in vitro, we transfected A549, and/or H1299 cells ($2\times10^5$) on six-well plates with a vector expressing *NPRL2* using lipofectamine. A549 and/or H1299 cells were cotransfected with 2 µg of *NPRL2* plasmid DNA (*pKGB2-NPRL2*) and 1 µg of *pcDNA3.1* vector (Invitrogen, Carlsbad, CA) containing a neomycin-resistant gene. Untransfected cells and cells transfected with the *pcDNA3.1* vector alone (1 µg) or with 2 µg of empty vector (*pKGB2-EV*) served as controls. Twenty-four hours after transfection, cells were harvested, stained with trypan blue, counted, and replated onto a 100 mm dish ($1\times10^4$ per dish) in triplicate. Cells were grown in RPMI 1640, supplemented with 10% FCS, and containing 400 µg/mL G418, for 2 wk. The G418-resistant colonies were counted after staining with crystal violet.

## Cell cycle assay

The cell cycle profiles of *NPRL2* stably expressing cells were determined by staining DNA with a fluorescent dye (PI/RNase staining buffer, BD Pharmingen, USA) according to the manufacturer's protocol, and measuring its intensity by flow cytometry (Attune NxT, Thermo Fisher Scientific, USA). Briefly, cells were seeded at $10^6$ cells in a 100 mm dish at the designated time. Cell pellets were suspended in ice-cold PBS, fixed with 70–80% ethanol, and stored at –20 °C overnight. The cells were washed twice with ice-cold PBS and stained with PI/RNase staining dye for 15 min at room temperature. The samples were analyzed by flow cytometry within an hour.

## Analysis of apoptosis by flow cytometry

Induction of apoptosis in tumor cells treated with *NPRL2* vector and *NPRL2* stably expressing cells (A549-*NPRL2*[++/++] and H1299-*NPRL2*[++/++]) with or without carboplatin treatments was analyzed with fluorescence-labeled Annexin V and PI staining Kit (BD Biosciences) by flow cytometry. Briefly, cells were plated on 60 mm dishes at $4\times10^5$ per dish for 24 hr and then treated with 4 µg of empty vector or *NPRL2*-expressing vector in the absence or presence of carboplatin. After 72 hr or indicated times mentioned in results, cells were harvested and fixed in 1% paraformaldehyde. Annexin V and PI-positive cells were detected and analyzed by FACS.

## Western blot analysis

Total protein was harvested using RIPA lysis buffer (Merck, Burlington, MA), and their concentrations were evaluated with BCA protein assay kit (Pierce, Rockford, IL, USA). Equal amounts of proteins

were separated by 8–15% SDS-PAGE gel, electro-transferred onto a Hybond ECL transfer membrane (Amersham Pharmacia, Piscataway, NJ), and blocked with 2–5% non-fat skim milk. Then, membranes were probed with specific primary antibodies at 1:1000 dilution for overnight at 4 °C, washed with PBS, and incubated with corresponding secondary antibodies at 1:2000 dilution at room temperature for 1 hr. The specific protein bands were visualized with an ECL advanced western blot analysis detection kit (GE Health Care Biosciences, NJ, USA). Antibodies used in this study were purchased from Cell Signaling (Beverly, MA): anti- AKT (CS#4691), p-AKT (Ser473) (CS#9271), p-mTOR (Ser2448) (CS#2971), p-MAPK (Thr202/Tyr204) (CS#4370), p-S6 (Ser240), p-S6 (Ser235), p-4E-BP1 (Ser65), p-PRAS40 (Thr246), p-GSK-3b (Ser9), and PARP. NPRL2 antibody was purchased from Abcam, Cambridge, MA. Monoclonal anti-β-actin (Sigma#A5449) was purchased from Sigma Aldrich (St Louis, MO).

## In-vivo tumor development and treatment in humanized mice

Anti-PD1 resistant *KRAS/STK11* mutant A549-luc lung metastasis was developed through intravenous injection of $1\times10^6$ A549-luc cells into NSG mice 6–8 wk post-CD34 engraftment. Tumor growth was measured by quantifying bioluminescence intensity with an IVIS small-animal imaging system (IVIS 200; Caliperls). Seven to ten days post A549 implantation in humanized mice, tumor growth and metastasis were confirmed by IVIS imaging. Then, animals were randomized into treatment and no-treatment groups based on tumor intensity and donor HLA type. Eight to ten mice per group from multiple umbilical cord blood donors were used. The treatment groups were untreated, *NPRL2*, pembrolizumab, and *NPRL2* + pembrolizumab. *NPRL2* (Genprex Inc) (25 µg/mouse) was injected intravenously every 3–4 d for six times; anti-PD1 agent pembrolizumab (Merck) (250 µg/mouse, intraperitoneally) every 3–4 d for three cycles. IVIS images of the mice were obtained once a week to monitor tumor progression based on bioluminescence intensity. H1299-luc subcutaneous tumors were developed by injecting 5 million/mouse H1299-luc cells into 6–8 wk post-humanized or non-humanized NSG mice subcutaneously. Tumor volume was measured periodically by calipers and bioluminescence imaging by IVIS 200 without knowledge of the treatment group. Subcutaneous tumor-bearing mice were randomized based on initial tumor volume (200 mm³) and donor HLA types. When tumor volume reaches 200 mm³, mice were randomized and treated with *NPRL2*, pembrolizumab, and its combination according to the doses mentioned above. Mice were monitored daily for side effects. Two perpendicular tumor diameters were measured twice per week, and tumor surface area was calculated according to a formula 1/2(Length × Width²).

## In-vivo tumor development and treatment in syngeneic mice

In the LLC2-luc syngeneic model, 6- to 8-wk-old female C57BL/6-Elite mice were injected (on day 0) with LLC2-luc cells ($1\times10^6$ cells/100 µL phosphate-buffered saline) subcutaneously in the right flank. On day 7 or indicated in the results, the mice were imaged using an IVIS 200 imaging platform (Caliper Life Sciences, Hopkinton, MA) and randomized into treatment groups based on tumor luminescence intensity. For therapeutic experiments, each treatment group consisted of 10 mice. The treatment groups were as follows: control (empty-vector nanovesicles, isotype antibody), anti–PD-1, *NPRL2* nanovesicles, and combination (*NPRL2* +anti–PD-1). Anti-PD-1 (catalog no. BE0146 [clone RMP1-14]) and InVivoPlus isotype Abs (catalog no. BE0089 [clone 2A3]) were purchased from Bio X Cell (West Lebanon, NH). Treatment schedules and dosages are shown schematically in figures. Briefly, 25 µg of *NPRL2* per mouse was injected intravenously every 3–4 d for six times, and 0.25 mg of anti–PD1 antibody was injected intraperitoneally (i.p.) every 4 d for three cycles. Mice were monitored daily for side effects. Two perpendicular tumor diameters were measured twice per week, and tumor surface area was calculated according to a formula 1/2(Length × Width²). In vivo imaging system (IVIS) images of the mice were obtained once a week to monitor tumor progression based on bioluminescence intensity. The mice were euthanized at 3–4 wk after tumor-cell injection and the tumors were harvested for immune histopathological analysis. For immune phenotyping experiments, mice were sacrificed within 3–5 d of last treatment, and fresh tumors were harvested for single-cell preparation for flow cytometry analysis.

## Immune cell depletion studies

InVivoPlus mAbs to mouse NK1.1 (BioXcell; catalog no. BP0036 [clone PK136]), to mouse CD8α (BioXcell; clone YTS 169.4), to mouse CD4 (BioXcell; Catalog no. BE0003-1; clone GK1.5) or an IgG control

were injected into the mice (100 µg, i.p.) every 3 d for four cycles beginning on day 0 after subcutaneous injection of cancer cells. NK-cells, CD4+ T, and CD8+ T-cell depletion status was monitored via flow cytometry analysis of splenocytes at various times to confirm the degree of depletion of those respective immune cells during treatment. CD4+ T, CD8+ T, and NK-cell depletion was studied in the LLC2-luc mouse model. The tumor growth in CD4+ T, CD8+ T, and NK-cell depleted mice given *NPRL2* or *NPRL2* and anti–PD1 treatment was monitored. Tumor volume was measured every other day for 3–4 wk.

## Tumor immune microenvironment analysis in humanized mice by multicolor flow cytometry

Erythrocytes in the peripheral blood were lysed with ACK lysis buffer (Fisher Scientific). Single-cell suspensions were prepared from fresh lung metastasis and spleen tissues using standard procedures. Several 10-color flow cytometry panels were used for immune profiling of both innate and adaptive immune populations in humanized mice and for evaluating immune response after treatment. Fluorochrome–conjugated monoclonal antibodies to the following human antigens were used: CD45-Alexa Fluor 700 (clone 2D1, HI30), CD45-phycoerythrin (PE; clone 2D1, HI30), CD3-PerCp/cy5.5 (clone HIT3a), CD19-PE-cyanine 7 (clone HIB19), CD8-allophycocyanin-cyanine 7 (clone RPA-T8, HIT8a), CD4-Pacific blue (clone OKT4), CD56-PE/BV510 (clone HCD56), CD69-FITC/APC/PE-Alexa Flour 610 (clone FN50; Thermo fisher), HLA-DR-PerCp/cy5.5 (clone LN3), CD33-PE (clone WM-53) (Thermo Fisher), CD11b-PE-Cy7 (clone 1CRF-44) (Thermo Fisher), Granzyme B-FITC (clone GB11), and IFN-γ-APC (clone 4 S.B3), CD103-Super bright 600 (Colne B-LY7; Thermo fisher), CD279 (PD-1)-Super Bright 702 (Clone J105; Thermo Fisher), CCR7-FITC (Clone G043H7), CD45RA-PE (Clone HI100), CD25-APC (clone CD25-4E3), Lin-FITC (Biolegend), CD163-APC (clone ebioGH1/61; Thermo fisher), CD11c-Pacific blue (clone Bu15; Thermo Fisher). A mouse CD45-FITC (clone 30-F11) antibody was used for gating out murine leukocytes. Most antibodies were purchased from BioLegend if otherwise mentioned. The dilutions of antibodies used in sample staining were followed according to the manufacturer's protocol. All samples were run on an Attune NxT flow cytometer (Thermo Fisher), and data were analyzed by Flow Jo and Kaluza software packages.

## Immune analysis in syngeneic tumors by flow cytometry

Single-cell suspensions from the spleen and LLC2 tumors were prepared within 3–5 d after the last treatment and stained for flow cytometry analysis. Multi-color panels for mouse samples were developed and optimized for use with a four lagers Attune NxT Flow Cytometer Research System (Thermo Fisher Scientific, USA). Mouse mAbs were purchased from BioLegend (San Diego, CA) unless otherwise mentioned. Single-cell suspensions were washed with fluorescence-activated cell sorting staining buffer, incubated with a mouse Fc receptor-binding inhibitor for 10 min, and stained with mAbs to CD45-Alexa Fluor 700 (clone 30-F11), CD3PerCP/Cy5.5 (clone 17A2), CD4-Briliant Violet 510 (clone GK1.5), CD8-FITC (clone 53–6.7), CD19-PE/Cy7 (clone 6D5), CD25-Briliant Violet 421 (clone PC61), CD49b-PE (clone DX5), CD62L-APC (clone MEL-14), and anti-mouse CD279 (PD-1) (clone 29 F.1A12) in 1 panel to analyze major immune populations. A myeloid panel was designed to analyze MDSCs using Brilliant Violet 510 CD11b (clone M1/70), APC/Cy7 I-A/I-E antibody (clone M5/114.15.2), FITC Ly-6G/Ly-6C (Gr-1) (clone RB6-8C5), PE/Cy7 CD68 (clone FA-11), Alexa Fluor 700 CD11c (N418), and Brilliant Violet 421 CD274 (PD-L1) (clone 10 F.9G2), anti-mouse CD86, anti-mouse CD40. The data were acquired using an Attune NxT Flow Cytometer (Beckman Coulter) and analyzed using FlowJo software version 10 (FlowJo, Ashland, OR).

## NanoString gene expression analysis in humanized mice system

Anti-PD1 resistant *KRAS/STK11* mutant A549 cells were injected into 6–8 wk post-humanized mice and lung metastasis was developed. Then *NPRL2* treatment was performed according to the protocol described above. Within 3–5 d of the last treatment, the mice were euthanized, and their harvested lung Mets were preserved in RNAlater solution. Total RNA was extracted from each tumor tissue using a Qiagen RNeasy Mini Kit. The RNA samples were submitted to the Genomic Core Facility at Baylor College of Medicine (Houston, TX) to run the NanoString panel. The NanoString nCounter[R] human PanCancer Immune Profiling panel (NanoString Technologies, Seattle, WA) which profiles 776 human genes related to specific immune-cell types and immune-cell functions was used for 12 tumor

samples subjected to anti–PD-1, *NPRL2*, combination (*NPRL2* +anti–PD-1), or empty-vector control treatment (three tumors per treatment group). RNA samples were subjected to quality control first to meet the standards for the NanoString experiment. Quality control-verified RNA was hybridized with the NanoString nCounter PanCancer Immune Profiling human panel code set and quantified using an nCounter Digital Analyzer at the Baylor College of Medicine Genomics Core Facility. The data were analyzed at the Bioinformatics Core Facility at MD Anderson.

## Statistical analysis

All experiments were designed and planned with biostatistician Dr. Jing Wang from the bioinformatics department at MD Anderson Cancer Center. For therapeutic studies in subcutaneous or metastasis tumors developed either in humanized or non-humanized or in syngeneic mouse model, generalized linear regression models were used to study the tumor growth over time. Statistical analyses were performed with GraphPad Prism 7 software. Tumor intensity change per time point was calculated as a relative level of tumor intensity change from baseline. Two-way ANOVA with interaction of treatment group and time point was performed to compare the difference of tumor intensity changes from baseline between each pair of the treatment groups at each time point. Means ± standard errors of the mean are shown in all graphs. The nonparametric Mann-Whitney U test was applied to compare cell numbers in different treatment groups. Differences of $p<0.05$, $p<0.01$, and $p<0.001$ were considered statistically significant.

Statistical analysis of flow cytometry data was done by general linear regression models to compare the data among the different treatment groups CONTRAST statement in PROC GENMOD procedure in SAS was used to compare the data between each pair of the treatment groups with treatment indicator in the models. Both nominal *P* values and multiple testing adjusted *P* values were reported. SAS version 9.2 and S-Plus version 8.04 are used for the computations for all analyses.

For Nano String data analysis, data normalization and statistical analysis were performed. Data generated by nCounter system is normalized prior to being used to quantify the gene profile and statistical analysis. The positive controls, housekeeping genes, and negative controls are used to adjust for sample preparation variation, background noise, and RNA content variation. The package *NanoStringNorm in* the R statistical language is used to pre-process the raw expression data. The linear model is used to evaluate the overall treatment effect and contrast is used to make pairwise comparisons of interest. The resulting p values are modeled using the beta-uniform mixture (BUM) model to determine a false discovery rate (FDR) cutoff and identify significantly differentially expressed genes. All statistical analyses are performed using R statistical software.

---

## Additional information

### Funding

| Funder | Grant reference number | Author |
| --- | --- | --- |
| National Cancer Institute | CA-016672 | Jack A Roth |
| Genprex Inc | Sponsor research agreement | Jack A Roth |

The funders had no role in study design, data collection and interpretation, or the decision to submit the work for publication.

### Author contributions

Ismail M Meraz, Conceptualization, Data curation, Formal analysis, Investigation, Methodology, Project administration, Resources, Software, Supervision, Validation, Visualization, Writing – original draft, Writing – review and editing; Mourad Majidi, Conceptualization, Project administration, Writing – original draft, Writing – review and editing; Renduo Song, Formal analysis, Investigation, Methodology, Software, Validation; Feng Meng, Formal analysis, Investigation, Methodology, Resources, Validation; Lihui Gao, Formal analysis, Validation, Writing – original draft; Qi Wang, Data curation, Investigation, Methodology, Validation; Jing Wang, Investigation, Methodology, Project administration, Validation; Elizabeth J Shpall, Data curation, Writing – review and editing; Jack A Roth, Conceptualization, Data

curation, Project administration, Funding acquisition, Resources, Software, Writing – original draft, Writing – review and editing

### Author ORCIDs
Ismail M Meraz ⬥ https://orcid.org/0000-0001-9109-5196

### Ethics

All animal experiments were carried out following approval by the University of Texas MD Anderson Cancer Center institutional review board under a IACUC protocol (#00000887) and were performed in accordance with the Guidelines for the Care and Use of Laboratory Animals published by the National Institutes of Health.

Reviewer #1 (Public review): https://doi.org/10.7554/eLife.98258.2.sa1
Reviewer #2 (Public review): https://doi.org/10.7554/eLife.98258.2.sa2
Reviewer #3 (Public review): https://doi.org/10.7554/eLife.98258.2.sa3
Author response https://doi.org/10.7554/eLife.98258.2.sa4

---

# Additional files

### Supplementary files
MDAR checklist

### Data availability
All data generated or analyzed during this study are included in the manuscript and source data files.

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
