## [Editor Report · eLife assessment]

This study provides a novel and promising NPRL2 gene therapy for enhanced immunotherapy response in a KRAS/STK11 mutant anti-PD1 resistant metastatic NSCLC humanized mouse model. Overall, the authors presented a large amount of **convincing** in vivo data to demonstrate that NPRL2 gene therapy induces antitumor activity through DC-mediated antigen presentation and cytotoxic immune cell activation. This work will be of interest and **useful** to medical biologists and oncologists in the research field of KRAS-mutant NSCLC.

---

## [Referee Report · Reviewer #1 (Public review)]

This study excellently complements the previous one by unveiling the properties of NPRL2 in augmenting the effect of immune checkpoint inhibitors such as pembrolizumab in KRAS mutant lung cancer models.

The following points should be clarified:

(1) In KRAS mutant cell lines with LKB1 co-mutations or deletions, such as A549 cells, does treatment with NPRL2 not increase the efficacy of immunotherapy? Is this correct? Similarly, does the delivery of NPRL2 only potentiate the effect of immunotherapy in KRAS mutant cell lines without associated LKB1 mutations?

(2) Do the authors analyze by western blot if NPRL2 influences or restores STING and LKB1 in the A549 cell line that lacks LKB1 and STING?

(3) Mechanistically, is there any explanation as to why NPRL2 delivery increases the efficacy of immunotherapy? Is there any effect on FUS or MYC?

(4) Is there any way to carry out a clinical study of systematically delivering NPRL2 in KRAS lung cancer patients?

---

## [Referee Report · Reviewer #2 (Public review)]

Summary:

NPRL2 gene therapy induces effective antitumor immunity in KRAS/STK11 mutant anti-PD1 resistant metastatic non-small cell lung cancer (NSCLC) in a humanized mouse model by Meraz et al investigated the antitumor immune responses to NPRL2 gene therapy in aPD1R / KRAS/STK11mt NSCLC in a humanized mouse model, and found that NPRL2 gene therapy induces antitumor activity on KRAS/STK11mt/aPD1R tumors through DC-mediated antigen presentation and cytotoxic immune cell activation.

Strengths:

The novelty of the study.

Weaknesses:

(1) The inconsistent effect of NPRL2 combined with pembrolizumab. Figure 2I-K, showed a similar tumor intensity in the NPRL2 group and combination group. However, NPRL2 combined with pembrolizumab was synergistic in the KRASwt/aPD1S H1299 tumors in Figure 4.

(2) The authors stated that NPRL2 combined with pembrolizumab was not synergistic in the KRAS/STK11mt/aPD1R tumors but was synergistic in the KRASwt/aPD1S H1299 tumors. How did the synergistic effect defined in the study, more details need to be provided here.

(3) Nearly all of the work was performed pre-clinically. Validation in the clinical setting would provide more strong evidence for the conclusion.

(4) Figure 5 and Figure 6 have the same legend. These 2 figures could be merged as a new one.

(5) Figure 5B & C, n=9 in the Figure 5B. However, the detail number in Figure 5C was less than 9.

---

## [Referee Report · Reviewer #3 (Public review)]

Summary:

NPRL2/TUSC4 is a tumor suppressor gene whose expression is reduced in many cancers including NSCLC. This study presents a novel finding on NPRL2 gene therapy, which induces antitumor activity on aPD1-resistant tumors. Since KRAS/STK11 mutant tumors were reported to be less benefited from ICIs, this study has potential clinical application value.

Strengths:

This work uncovers the advantage of NPRL2 gene therapy by using humanized models and multiple cell lines. Moreover, via immune cell depletion studies, the mechanism of NPRL2 gene therapy has focused on dendritic cells and CD8+T cells.

Weaknesses:

A major concern would be the lack of systematic, and logical rigor. This work did not present a link between apoptosis and antigen presenting induced by NPRL2 restoration. There is no evidence proving that the PI3K/AKT/mTOR signaling pathway is related to antigen presenting, which is the major reason of NPRL2 induced antitumor response. Therefore, the two parts may not support each other logically.

---

## [Author Response]

**Reviewer #1 (Public Review):**
This study excellently complements the previous one by unveiling the properties of NPRL2 in augmenting the effect of immune checkpoint inhibitors such as pembrolizumab in KRAS mutant lung cancer models.The following points should be clarified:(1) In KRAS mutant cell lines with LKB1 co-mutations or deletions, such as A549 cells, does treatment with NPRL2 not increase the efficacy of immunotherapy? Is this correct? Similarly, does the delivery of NPRL2 only potentiate the effect of immunotherapy in KRAS mutant cell lines without associated LKB1 mutations?

NPRL2, when used as a single-agent immunotherapy, induces robust antitumor activity in immunotherapy-resistant (aPD1R) KRAS mutant models, such as A549 tumors (KRASmt/LKB1mt/aPD1R) and LLC2 (KRASmt/aPD1R), where immunotherapy is ineffective regardless of LKB1 co-mutation or deletion status. The antitumor effect of NPRL2 combined with aPD1 immunotherapy was not significantly different from NPRL2 alone in immunotherapy-resistant models but was significantly greater than immunotherapy alone. However, a synergistic antitumor effect was observed with NPRL2 and aPD1 immunotherapy in KRAS wild-type and immunotherapy-moderately-responsive models, such as H1299 (KRASwt/aPD1S).

(2) Do the authors analyze by western blot if NPRL2 influences or restores STING and LKB1 in the A549 cell line that lacks LKB1 and STING?

NPRL2 induces antitumor immunity on Kras mutant, aPD1 resistant models regardless of LKB1 co-mutations or deletions, however, it would be interesting to look into the effect of NPRL2 on the STING pathway in this LKB1 deleted A549 cell line.

(3) Mechanistically, is there any explanation as to why NPRL2 delivery increases the efficacy of immunotherapy? Is there any effect on FUS or MYC?

NPRL2 is a multifunctional tumor suppressor gene that is downregulated or absent in many cancers. NPRL2 has been shown to induce apoptosis, inhibit cell proliferation, and cause cell cycle arrest in various cancer types. Compelling evidence highlights the critical role of NPRL2 in causing DNA damage and double-strand breaks, which can trigger dendritic cell (DC) activation, antigen presentation, and priming of tumor-specific CD8+ T cells in the tumor microenvironment (TME). Our data indicate that NPRL2 treatment is associated with the induction of DC activation and maturation.

The cellular mechanism of NPRL2 suggests that NPRL2-mediated antitumor immunity depends on the presence of CD4+ T cells, CD8+ T cells, and macrophages. Interestingly, the expression of FUS1, another tumor suppressor gene, was mostly absent or severely downregulated in most non-small cell lung cancers (NSCLC) and was unaffected by NPRL2 treatment. While MYC expression was not assessed in this study, it remains an area of interest for future research.

(4) Is there any way to carry out a clinical study of systematically delivering NPRL2 in KRAS lung cancer patients?

In this preclinical study, a clinical-grade DOTAP-NPRL2 formulation was prepared, utilizing NPRL2 encapsulated within nanovesicles for delivery. Based on the promising preclinical data, a phase I clinical trial will be initiated to evaluate the safety and efficacy of this formulation.

**Reviewer #2 (Public Review):**
Summary:NPRL2 gene therapy induces effective antitumor immunity in KRAS/STK11 mutant anti-PD1 resistant metastatic non-small cell lung cancer (NSCLC) in a humanized mouse model by Meraz et al investigated the antitumor immune responses to NPRL2 gene therapy in aPD1R / KRAS/STK11mt NSCLC in a humanized mouse model, and found that NPRL2 gene therapy induces antitumor activity on KRAS/STK11mt/aPD1R tumors through DC-mediated antigen presentation and cytotoxic immune cell activation.Strengths:The novelty of the study.Weaknesses:(1) The inconsistent effect of NPRL2 combined with pembrolizumab. Figure 2I-K, showed a similar tumor intensity in the NPRL2 group and combination group. However, NPRL2 combined with pembrolizumab was synergistic in the KRASwt/aPD1S H1299 tumors in Figure 4.

NPRL2, as a single agent immunogen therapy, induces robust antitumor activity on both immunotherapy-resistant (aPD1R) KRAS mutant models, such as A549 tumors (KRASmt/LKB1mt/aPD1R) and LLC2 (KRASmt/aPD1R) and immunotherapy sensitive model such as H1299 (KRASwt/aPD1S) where immunotherapy was ineffective or limitedly effective. A synergistic antitumor effect of NPRL2 and Pembrolizumab combination was found only in immunotherapy moderately responsive models, not in immunotherapy resistant models where PD-1/PD-L1 signaling is impaired shown in Figure 1A.

(2) The authors stated that NPRL2 combined with pembrolizumab was not synergistic in the KRAS/STK11mt/aPD1R tumors but was synergistic in the KRASwt/aPD1S H1299 tumors. How did the synergistic effect defined in the study, more details need to be provided here.

Our biostatistician used generalized linear regression models to study the tumor growth over time. Two-way ANOVA with the interaction of treatment group and time point was performed to compare the difference of tumor intensity changes from baseline between each pair of the treatment groups at each time point. The nonparametric Mann-Whitney U test was applied to compare significance in different treatment groups. Differences of P < 0.05, P < 0.01, and P < 0.001 were considered statistically significant. When the combination antitumor effect of NPRL2 and pembrolizumab was found to be statistically significant compared to both single-agent effects synergy was confirmed using the method of Huang et al.

Huang L, Wang J, Fang B, Meric-Bernstam F, Roth JA, Ha MJ. CombPDX: a unified statistical framework for evaluating drug synergism in patient-derived xenografts. Sci Rep 12(1):12984, 7/2022. e-Pub 7/2022. PMCID: PMC9338066.

(3) Nearly all of the work was performed pre-clinically. Validation in the clinical setting would provide more strong evidence for the conclusion.

In this preclinical study, a clinical-grade DOTAP-NPRL2 formulation was prepared, utilizing NPRL2 encapsulated within nanovesicles for delivery. Based on the promising preclinical data, a phase I clinical trial will be initiated to evaluate the safety and efficacy of this formulation.

(4) Figure 5 and Figure 6 have the same legend. These 2 figures could be merged as a new one.

Agreed.

(5) Figure 5B & C, n=9 in the Figure 5B. However, the detail number in Figure 5C was less than 9.

At least n=7-9 mice/group are shown in the figure 5C. We will revise accordingly.

**Reviewer #3 (Public Review):**
Summary:NPRL2/TUSC4 is a tumor suppressor gene whose expression is reduced in many cancers including NSCLC. This study presents a novel finding on NPRL2 gene therapy, which induces antitumor activity on aPD1-resistant tumors. Since KRAS/STK11 mutant tumors were reported to be less benefited from ICIs, this study has potential clinical application value.Strengths:This work uncovers the advantage of NPRL2 gene therapy by using humanized models and multiple cell lines. Moreover, via immune cell depletion studies, the mechanism of NPRL2 gene therapy has focused on dendritic cells and CD8+T cells.Weaknesses:A major concern would be the lack of systematic, and logical rigor. This work did not present a link between apoptosis and antigen presenting induced by NPRL2 restoration. There is no evidence proving that the PI3K/AKT/mTOR signaling pathway is related to antigen presenting, which is the major reason of NPRL2 induced antitumor response. Therefore, the two parts may not support each other logically.

Thank you for your review and comments. We agree that future studies are necessary to establish a direct link between apoptosis and antigen presentation induced by NPRL2 restoration, as well as NPRL2-mediated downregulation of PI3K/AKT/mTOR signaling and its direct effect on antigen presentation. Although NPRL2 restoration directly induced apoptosis in several cell lines shown in Figure 1C and Figure 8Q and significantly increased the number of antigen-presenting DC cells in the tumor microenvironment upon NPRL2 treatment or NPRL2 restoration. Similarly, NPRL2 restoration downregulated the PI3K/AKT/mTOR pathway, which was associated with increased antitumor immunity.